# Bayesian Optimization with Informative Covariance

**Afonso Eduardo**
*School of Informatics*
*University of Edinburgh*

*afonso.eduardo@ed.ac.uk*

**Michael U. Gutmann**
*School of Informatics*
*University of Edinburgh*

*michael.gutmann@ed.ac.uk*

## Abstract

Bayesian optimization is a methodology for global optimization of unknown and expensive objectives. It combines a surrogate Bayesian regression model with an acquisition function to decide where to evaluate the objective. Typical regression models are given by Gaussian processes with stationary covariance functions. However, these functions are unable to express prior input-dependent information, including possible locations of the optimum. The ubiquity of stationary models has led to the common practice of exploiting prior information via informative mean functions. In this paper, we highlight that these models can perform poorly, especially in high dimensions. We propose novel informative covariance functions for optimization, leveraging nonstationarity to encode preferences for certain regions of the search space and adaptively promote local exploration during optimization. We demonstrate that the proposed functions can increase the sample efficiency of Bayesian optimization in high dimensions, even under weak prior information.

## 1 Introduction

Bayesian optimization (BO) is a methodology for global function optimization that has become popular since the work of Jones et al. (1998). Other influential works date back to Kushner (1962) and Močkus (1975). BO has been applied to a broad range of problems (see e.g. Shahriari et al., 2016), including chemical design (Gómez-Bombarelli et al., 2018), neural architecture search (White et al., 2021) and simulation-based inference (Gutmann et al., 2016; Cranmer et al., 2020).

As a Bayesian approach, the idea is to place a prior over the objective function, typically a Gaussian process (GP) prior characterized by a mean and covariance function. The prior, combined with an observation model and past function evaluations (observations), allows the computation of a posterior predictive distribution. The next acquisition is determined by the posterior predictive and a utility function, treating the optimization as a decision problem.

Standard BO is generally effective in relatively low-dimensional problems. However, as dimensionality increases, statistical and computational problems become more noticeable. In particular, the curse of dimensionality manifests itself while learning the probabilistic surrogate of the objective function and during the acquisition process. Indeed, as training data become sparser, the estimation of an accurate surrogate becomes more difficult and the reliance on extrapolation increases. In this context, previous work has relied on certain structural assumptions (Binois & Wycoff, 2022), namely low effective dimensionality (Garnett et al., 2014; Li et al., 2016; Wang et al., 2016; Letham et al., 2020; Moriconi et al., 2020; Raponi et al., 2020; Eriksson & Jankowiak, 2021; Grosnit et al., 2021) and additivity (Kandasamy et al., 2015; Gardner et al., 2017; Mutný & Krause, 2018; Rolland et al., 2018; Wang et al., 2018). Local methods based on space-partitioning schemes and trust regions have also been proposed (Assael et al., 2014; McLeod et al., 2018; Eriksson et al., 2019; Wang et al., 2020b; Diouane et al., 2021). Notably, these methods employ stationary covariance functions.

By construction, stationary covariance functions are translation invariant and thus unable to express spatially-varying information. As shown in Figure 1, stationary models can lead to poor optimization of

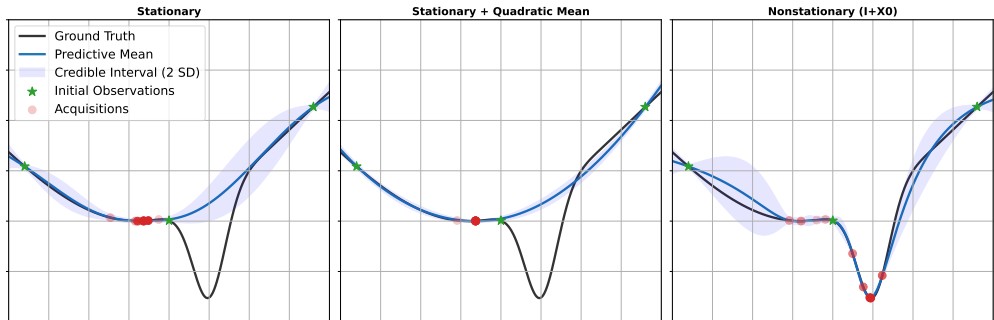

Figure 1: Overconfident models are unable to capture the ground truth, causing BO to yield suboptimal results. The stationary model with an informative quadratic mean fits the trend too well, resulting in small residuals. These small residuals lead to a smaller prior variance and a longer lengthscale that, in turn, decrease epistemic uncertainty. Stationarity posits that these hyperparameters are constant, but this assumption does not hold for functions with irregular bumps. The proposed informative priors, being nonstationary, allow for the specification of spatially-varying properties.

relatively simple one-dimensional objectives, suggesting issues in higher-dimensional domains. A common practice is to include input-dependent information via informative mean functions (see e.g. Snoek et al., 2015; Acerbi, 2019; Järvenpää et al., 2021). However, as demonstrated in Figure 1, this practice does not necessarily alleviate the limitations of stationary covariance functions. In contrast, nonstationarity allows for the incorporation of input-dependent information directly in covariance functions. The crux lies in the design of flexible and scalable informative GP priors.

In the remainder of this paper, after we introduce the background in Section 2, we lay the foundation for our methodology by discussing the advantages of second-order nonstationary for optimization in Section 3. focusing on spatially-varying prior variance, lengthscales and high-dimensional domains. This includes a discussion of regret bounds for covariance functions with spatially-varying parameters, which we have not previously found in the literature. We then make the following contributions:

- In Section 4, we design informative covariance functions that leverage nonstationarity to incorporate information about promising points. Our GP priors encode preferences for certain regions and allow for multiscale exploration during optimization. To our knowledge, we are the first to consider informative GP priors for optimization with spatially-varying prior variances and lengthscales, both induced by a shaping function that captures information about promising points.

- In Section 5, we empirically demonstrate that the proposed methodology can increase the sample efficiency of BO in high-dimensional domains, even under weak prior information. Additionally, we show that it can complement existing methodologies, including GP models with informative mean functions, trust-region optimization and belief-augmented acquisition functions.

Section 6 discusses related work and Section 7 concludes the paper.

## 2 Background

### 2.1 Bayesian Optimization

In global optimization, the goal is to find the global minimizer of an objective function $f \colon \mathbb{X} \to \mathbb{Y} \subseteq \mathbb{R}$,

$$\boldsymbol{x}^{\star} = \underset{\boldsymbol{x} \in \mathbb{X}}{\arg\min} \; f(\boldsymbol{x}), \tag{1}$$

where the domain $\mathbb{X} \subseteq \mathbb{R}^D$, for objectives of $D$ variables, and $f$ is a function whose expression and gradient are typically unavailable. Evaluation is possible at any point $\boldsymbol{x}_i \in \mathbb{X}$, yielding the observation $y_i = f(\boldsymbol{x}_i)$, but its cost is high enough to justify the use of sample-efficient methodologies that aim to minimize the number of evaluations by careful selection. In this context, Bayesian optimization (BO) emerges as a competitive approach. The idea is to train a Bayesian regression model $\mathcal{M}$ that explains the evidence collected up to step $n$, $\mathcal{D}_n = \{(\boldsymbol{x}_i, y_i)\}_{i \leq n}$, and provides a distribution over possible functions. Given this probabilistic

**Algorithm 1** Bayesian Optimization (BO)

---
**Input:** objective function $f$, acquisition function $\alpha$, statistical model $\mathcal{M}$, initial evidence $\mathcal{D}_{n_0}$
**repeat**
    $\boldsymbol{x}_{n+1} = \arg\max \alpha(\boldsymbol{x} \mid \mathcal{D}_n, \mathcal{M})$                            ▷ Find best candidate
    $y_{n+1} = f(\boldsymbol{x}_{n+1})$                                             ▷ Evaluate candidate
    $\mathcal{D}_{n+1} = \mathcal{D}_n \cup \{(\boldsymbol{x}_{n+1}, y_{n+1})\}$                ▷ Update evidence set
**until** stopping condition is met

---

surrogate, an acquisition function $\alpha$ assigns a utility to each candidate point, guiding the selection of the next acquisition, i.e., the next point to evaluate. The procedure is outlined in Algorithm 1, where the stopping condition may be based on a maximum evaluation budget and a tolerance. For more details on BO, see e.g. Shahriari et al. (2016), Frazier (2018) and Greenhill et al. (2020).

## 2.2 Gaussian Process Regression

BO is traditionally paired with Gaussian process (GP) models. For a given GP prior $f \sim GP(m_{\boldsymbol{\theta}}, C_{\boldsymbol{\theta}})$ with mean function $m_{\boldsymbol{\theta}} \colon \mathbb{X} \to \mathbb{Y}$, covariance function $C_{\boldsymbol{\theta}} \colon \mathbb{X} \times \mathbb{X} \to \mathbb{R}^+$, and an additive Gaussian observation model $y = f(\boldsymbol{x}) + \epsilon$, $\epsilon \sim \mathcal{N}(0, \sigma_y^2)$, the univariate posterior predictive distribution of the latent $f$ at test input $\boldsymbol{x}$ is Gaussian with mean $m_n$ and variance $\sigma_n^2$,

$$m_n(\boldsymbol{x}) = m_{\boldsymbol{\theta}^\star}(\boldsymbol{x}) + \boldsymbol{c}_n(\boldsymbol{x})^\intercal [\boldsymbol{C}_n + \sigma_y^2 \boldsymbol{I}]^{-1}(\boldsymbol{y}_n - \boldsymbol{m}_n), \tag{2}$$

$$\sigma_n^2(\boldsymbol{x}) = C_n(\boldsymbol{x}, \boldsymbol{x}), \tag{3}$$

$$C_n(\boldsymbol{x}_i, \boldsymbol{x}_j) = C_{\boldsymbol{\theta}^\star}(\boldsymbol{x}_i, \boldsymbol{x}_j) - \boldsymbol{c}_n(\boldsymbol{x}_i)^\intercal [\boldsymbol{C}_n + \sigma_y^2 \boldsymbol{I}]^{-1} \boldsymbol{c}_n(\boldsymbol{x}_j), \tag{4}$$

$$\boldsymbol{c}_n(\boldsymbol{x}) = (C_{\boldsymbol{\theta}^\star}(\boldsymbol{x}, \boldsymbol{x}_1), \ldots, C_{\boldsymbol{\theta}^\star}(\boldsymbol{x}, \boldsymbol{x}_n))^\intercal, \tag{5}$$

where the entries in the Gram matrix $\boldsymbol{C}_n$ are $[\boldsymbol{C}_n]_{ij} = C_{\boldsymbol{\theta}^\star}(\boldsymbol{x}_i, \boldsymbol{x}_j)$, and the mean vector $\boldsymbol{m}_n$ has entries $[\boldsymbol{m}_n]_i = m_{\boldsymbol{\theta}^\star}(\boldsymbol{x}_i)$, $\forall (\boldsymbol{x}_i, y_i) \in \mathcal{D}_n$. Hyperparameters $\boldsymbol{\theta}^\star$ can, e.g., be fixed a priori or learned via empirical Bayes, by introducing a hyperprior $p(\boldsymbol{\theta})$ and maximizing the unnormalized posterior $p(\boldsymbol{\theta}|\mathcal{D}_n) \propto p(\boldsymbol{y}_n|X_n, \boldsymbol{\theta})p(\boldsymbol{\theta})$, where the marginal likelihood is $p(\boldsymbol{y}_n|X_n, \boldsymbol{\theta}) = \mathcal{N}(\boldsymbol{y}_n; \boldsymbol{m}_n, \boldsymbol{C}_n + \sigma_y^2 \boldsymbol{I})$, $X_n = \{\boldsymbol{x}_i\}_{i \leq n}$. For a comprehensive introduction to GPs, see Rasmussen & Williams (2005).

## 2.3 Stationary Covariance Functions

Most popular covariance functions are translation invariant, depending only on the relative position of the input points. These functions compute $\mathrm{Cov}(f(\boldsymbol{x}_i), f(\boldsymbol{x}_j))$ based on a translation-invariant distance between $\boldsymbol{x}_i$ and $\boldsymbol{x}_j$. Valid covariance functions must be symmetric and positive definite. In this family, we find the squared exponential or Gaussian covariance,

$$C_{\mathrm{G}}(\boldsymbol{x}_i, \boldsymbol{x}_j) = \sigma_0^2 \exp(-1/2 \, d_{\mathrm{M}}^2(\boldsymbol{x}_i, \boldsymbol{x}_j)), \quad d_{\mathrm{M}}(\boldsymbol{x}_i, \boldsymbol{x}_j) = \sqrt{(\boldsymbol{x}_i - \boldsymbol{x}_j)^\intercal \boldsymbol{\Lambda}^{-1}(\boldsymbol{x}_i - \boldsymbol{x}_j)}, \tag{6}$$

where $\sigma_0^2$ is the *prior variance* and $d_{\mathrm{M}}$ is the Mahalanobis distance function with matrix $\boldsymbol{\Lambda}$. For simplicity and scalability, it is customary to constrain the matrix to be diagonal with entries $[\boldsymbol{\Lambda}]_{dd} = \lambda_d^2 \in \mathbb{R}^+$, $d \in \{1, ..., D\}$, known as (squared) *lengthscales*. In this case, the Mahalanobis distance turns into the weighted Euclidean distance, which we denote by $d_{\mathrm{WE}}$. These lengthscales control the exponential decay of correlation. For instance, a short lengthscale leads to sample paths, or realizations, that vary more rapidly.

The Gaussian covariance, being infinitely differentiable, may generate unrealistically smooth sample paths. A more general class of functions is generated, or reproduced, by the Matérn covariance function which controls the differentiability of the process via a smoothness parameter $\nu$, recovering $C_{\mathrm{G}}$ when $\nu \to \infty$. For optimization purposes, good empirical results have been found with $\nu = 5/2$ (Snoek et al., 2012), for which

$$C_{\mathrm{M}}(\boldsymbol{x}_i, \boldsymbol{x}_j) = \sigma_0^2 \left(1 + \sqrt{5} d_{\mathrm{M}}(\boldsymbol{x}_i, \boldsymbol{x}_j) + \frac{5}{3} d_{\mathrm{M}}^2(\boldsymbol{x}_i, \boldsymbol{x}_j)\right) \exp\left(-\sqrt{5} d_{\mathrm{M}}(\boldsymbol{x}_i, \boldsymbol{x}_j)\right). \tag{7}$$

An attractive property of the Gaussian and mixtures thereof, such as Matérn, is their proven universality (Micchelli et al., 2006). Asymptotically, these covariance functions, or kernels, are capable of approximating any continuous function on any compact subset of the domain. In practice, however, other covariance functions may be preferable in the small sample regime, especially if the elicited prior includes relevant information, or structure, that accelerates the learning process.

## 2.4 Acquisition Functions

While many rules to determine promising candidate solutions have been proposed (see e.g. Wilson et al., 2018; Neiswanger et al., 2022), those with an analytical expression remain the most popular. Such acquisition functions can be evaluated without Monte Carlo or quadrature approximations. Given a Bayesian model, myopic acquisition functions $\alpha$ suggest one acquisition at a time, depending only on the univariate posterior predictive distribution with mean $m_n$ and variance $v_n$, i.e., $\alpha(\boldsymbol{x} \mid \mathcal{D}_n, \mathcal{M}) = \alpha(\boldsymbol{x} \mid m_n, v_n)$. In this context, a popular choice is the Lower Confidence Bound (LCB) criterion (Srinivas et al., 2010),[1]

$$\text{LCB}(\boldsymbol{x}) = m_n(\boldsymbol{x}) - \beta_n \sigma_n(\boldsymbol{x}), \tag{8}$$

where the factor $\beta_n$ is related to a confidence level, balancing exploitation and exploration. Another popular alternative is the Expected Improvement (EI), which has been found to be more effective than LCB when properties of the objective function, e.g. norm bounds, are unknown (Snoek et al., 2012; Chowdhury & Gopalan, 2017; Merrill et al., 2021). This acquisition function is defined as

$$\text{EI}(\boldsymbol{x}) = \mathbb{E}_{p(f(\boldsymbol{x})|\boldsymbol{x}, \mathcal{D}_n)}[\text{I}(\boldsymbol{x})] = \sigma_n(\boldsymbol{x})\tau(z(\boldsymbol{x})), \tag{9}$$

$$\tau(z(\boldsymbol{x})) = z(\boldsymbol{x})F_{\mathcal{N}}(z(\boldsymbol{x})) + \mathcal{N}(z(\boldsymbol{x}); 0, 1), \quad z(\boldsymbol{x}) = (f(\boldsymbol{x}_{\text{best}}) - m_n(\boldsymbol{x}))/\sigma_n(\boldsymbol{x}), \tag{10}$$

where $\text{I}(\boldsymbol{x}) = \max(0, f(\boldsymbol{x}_{\text{best}}) - f(\boldsymbol{x}))$ is the improvement over the incumbent $f(\boldsymbol{x}_{\text{best}})$ and $F_{\mathcal{N}}$ is the standard Normal cumulative distribution function (CDF). For noisy functions, $f(\boldsymbol{x}_{\text{best}})$ is replaced by the minimum posterior predictive mean, $m_n^- = \min_{\boldsymbol{x}} m_n(\boldsymbol{x})$.

## 2.5 Regret

In the context of optimization, there are several useful metrics to measure performance. The instantaneous regret is the loss incurred at step $n$, $r_n = f(\boldsymbol{x}_n) - f(\boldsymbol{x}^\star)$, where $\boldsymbol{x}_n$ is the acquisition and $\boldsymbol{x}^\star$ is the global minimizer of $f$. The simple regret is the minimum instantaneous regret incurred up to step $n$, $s_n = \min_{t \leq n} r_t$, or, equivalently, the regret incurred by the incumbent solution. The cumulative regret is defined as the sum of instantaneous regret over $N$ steps, $R_N = \sum_{n \leq N} r_n$. If the cumulative regret grows at a sublinear rate, then the average regret goes to zero, $\lim_{N \to \infty} R_N/N = 0$, which is also known as no regret. This property implies that simple regret goes to zero, ensuring that the global optimum is asymptotically found. For more details on regret bounds, see e.g. Vakili et al. (2021) and Gupta et al. (2022).

# 3 Benefits of Second-Order Nonstationarity for Optimization

We now turn our attention to GP priors with spatially-varying properties, namely prior variance and length-scale, which provide the foundations for the informative covariance functions in Section 4. While an analysis in terms of cumulative regret is customary, we focus on the instantaneous regret because we are interested in the relatively small sample regime, for which the asymptotics are not relevant. Furthermore, by demonstrating that the instantaneous regrets $r_n$ are smaller (or larger), it follows that the cumulative regret $R_N$ is also smaller (or larger).

**Spatially-varying prior variance** The prior variance $\sigma_0^2$ quantifies the (epistemic) uncertainty of the surrogate model of $f$ before observing data. Relatedly, posterior predictive variances $\sigma_n^2(\boldsymbol{x})$, defined in Equation 3, determine the mutual information between observations $\boldsymbol{y}_N$ and the objective function $f$, $\text{MI}_N(\boldsymbol{y}_N; f) = \frac{1}{2} \sum_{n=1}^{N} \log\left(1 + \sigma_y^{-2}\sigma_{n-1}^2(\boldsymbol{x}_n)\right)$, see e.g. (Srinivas et al., 2010, Lemma 5.3). Then, for stationary covariance functions, any point $\boldsymbol{x}_n$ where $\boldsymbol{c}_{n-1}(\boldsymbol{x}_n) \approx \boldsymbol{0}$ provides equal information about $f$ because

---

[1] The LCB needs to be minimized and thus corresponds to a negative acquisition function $\alpha$.

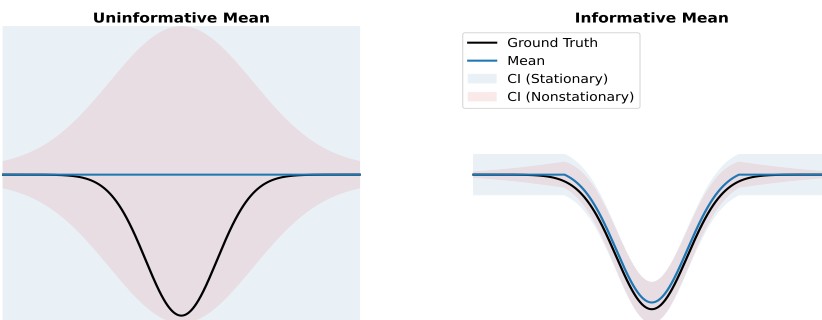

Figure 2: Narrower credible intervals (CI) can be obtained with a spatially-varying prior variance. Typically, GP priors in BO are uninformative, characterized by constant prior mean functions and stationary covariance functions that posit a constant prior variance. Then, for popular acquisition functions, all candidates are considered equally good a priori. Conversely, spatially-varying prior variances can encode preferences. In this example, points near the center are more informative a priori.

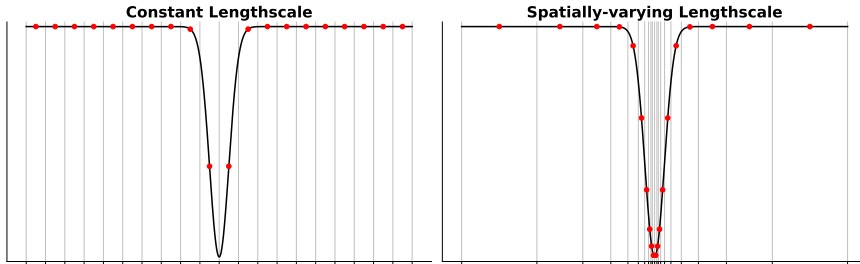

Figure 3: Given the same evaluation budget ($N = 20$), a model with spatially-varying lengthscales can better approximate the function than one with a constant lengthscale. While the latter must set a shorter global lengthscale, a multiscale approach can locally assign shorter lengthscales.

the predictive variance is approximately constant, $\sigma_{n-1}^2(\boldsymbol{x}_n) \approx \sigma_0^2$. Conversely, surrogate models of $f$ with a spatially-varying prior variance are spatially informative, even in the absence of neighboring observations, in which case $\sigma_{n-1}^2(\boldsymbol{x}_n) \approx \sigma_0^2(\boldsymbol{x}_n)$.

In terms of optimization, a spatially-varying prior variance can express a priori preferences for certain regions, as illustrated in Figure 2. Notably, compared to the stationary case, better optimization performance can be achieved: For popular acquisition functions, the worst-case instantaneous regret is proportional to the predictive standard deviation (Lemma A.1), provided that the interval $|f(\boldsymbol{x}) - m_n(\boldsymbol{x})| \leq \beta_n \sigma_n(\boldsymbol{x})$ holds for some factor $\beta_n$. Then, recall that the predictive variance is upper bounded by the prior variance, $\sigma_n^2(\boldsymbol{x}) \leq \sigma_0^2(\boldsymbol{x})$. While the latter is constant for stationary covariance functions, covariance functions with spatially-varying prior variances allow for narrower intervals and, in turn, tighter regret bounds. As a caveat, standard regret analysis assumes that 1) intervals include the objective function and 2) global optima of acquisition functions can be found, but in practice these assumptions do not necessarily hold.

**Spatially-varying lengthscale** In terms of sample efficiency, stationary covariance functions are ill-suited for approximating objectives with spatially-varying properties. Intuitively, a function that varies rapidly in one region may be characterized by short lengthscales, but is, otherwise, more efficiently described by longer lengthscales. In order to capture the finest details of such a function, a stationary covariance would require the shortest possible lengthscale, resulting in additional observations to approximate the function and ultimately find the optimum. In contrast, a multiscale approach only requires more neighboring observations where the local lengthscales are shorter (Figure 3).

In more detail, let the objective be a realization from a locally stationary process, $f(\boldsymbol{x}) = \sum_{m \leq M} \mathbf{1}_{\boldsymbol{x} \in \mathbb{X}_m} f_m(\boldsymbol{x})$ with disjoint $\mathbb{X}_m \subseteq \mathbb{X}$, $m = \{1, \ldots, M\}$, $M \in \mathbb{N}^+$. Each component is a member of a class of functions forming the Hilbert space $\mathcal{H}_{C_m}(\mathbb{X}_m)$ with reproducing stationary kernel $C_m$ and bounded norm $\|f\|_{\mathcal{H}_{C_m}} \leq B_m$. If $f$ is to be estimated with a stationary covariance function $C$, then it must reproduce the space $\mathcal{H}_C(\mathbb{X}) \supseteq \bigcup_{m \leq M} \mathcal{H}_{C_m}(\mathbb{X}_m)$ with $\boldsymbol{x} \in \mathbb{X} = \bigcup_{m \leq M} \mathbb{X}_m$ and norm bound $B \geq \sqrt{\sum_m B_m^2}$, by (Aronszajn, 1950, Theorem §6). This corresponds to a search in the space of more complex functions, for which more

observations are required because the global lengthscale must be set as the shortest from $\{C_m\}_{m \leq M}$. Indeed, for shorter lengthscales, $\lambda' \leq \lambda$, we have larger posterior predictive variances, $\sigma_{n,\lambda'}^2(\boldsymbol{x}) \geq \sigma_{n,\lambda}^2(\boldsymbol{x})$, which lead to greater information gains $\mathrm{MI}_{n,\lambda'} \geq \mathrm{MI}_{n,\lambda}$, and larger norms, $\|f\|_{\mathcal{H}_{C_{\lambda'}}} \geq \|f\|_{\mathcal{H}_{C_\lambda}}$, by (Bull, 2011, Lemma 4), contributing to a larger $\beta_{n,\lambda'} \geq \beta_{n,\lambda}$ because $\beta_n = \sup_{f \in \mathcal{H}_C, \mathrm{MI}_n} \|f\|_{\mathcal{H}_C} + \sigma_y \sqrt{2(\mathrm{MI}_n + 1 - \log \delta)}$, $\delta \in (0,1]$, by (Chowdhury & Gopalan, 2017, Theorem 2). Then, by Lemma A.1, wider intervals $|f(\boldsymbol{x}) - m_n(\boldsymbol{x})| \leq \beta_n \sigma_n(\boldsymbol{x})$ lead to larger regret bounds, and hence worse performance.

The argument above provides additional justification for spatially-varying prior variances, in which case each $C_m$ is characterized by different but constant $\sigma_{0,m}^2$. Then, the stationary covariance function must set $\sigma_0^2 = \max_m \sigma_{0,m}^2$, leading again to wider intervals, and hence worse performance by Lemma A.1. Finally, as $M$ grows, it becomes more challenging to estimate and maintain $M$ local models. Nonstationary approaches that do not rely on space partitions can avoid these computational and statistical problems. Notably, in the limit, each subset is a singleton, $\mathbb{X}_m = \{\boldsymbol{x}_m\}$, $\forall \boldsymbol{x}_m \in \mathbb{X}$, and the restriction $f|\mathbb{X}_m$ is reproduced by $C_m(\boldsymbol{x}_i, \boldsymbol{x}_j) = \delta_{im}\delta_{jm}$ with norm $\|f\|_{\mathcal{H}_{C_m}} = |f(\boldsymbol{x}_m)|$. For an uninformative prior mean, $m_0(\boldsymbol{x}) = 0$, the narrowest interval, before observing data, $|f(\boldsymbol{x}_m) - m_0(\boldsymbol{x}_m)| \leq \beta_{0,m}\sigma_{0,m}$, is obtained with a spatially-varying $\beta_0(\boldsymbol{x}) = |f(\boldsymbol{x})|$ and constant $\sigma_0 = 1$, or, equivalently, with $\beta_0 = 1$ and spatially-varying prior standard deviation $\sigma_0(\boldsymbol{x}) = |f(\boldsymbol{x})|$. Thus, the optimal prior variance depends on the shape of the objective function, as previously suggested by Figure 2.

**High-dimensional domains** In high-dimensional Euclidean spaces, most of the volume is on the boundary, and for stationary GPs, epistemic uncertainty, measured by the posterior predictive variance, increases with the distance to the training data, tilting the acquisition step toward blind exploration of the boundary, typically faces of a hypercube (Binois & Wycoff, 2022). This problem is known as the *boundary issue* (Swersky, 2017; Oh et al., 2018).

Moreover, uninformative priors provide no practical finite-time convergence guarantees. Begin by recalling that the motivation behind BO is the optimization of expensive functions, which puts a practical constraint on the size of the evaluation budget. However, unless the total budget $N$ increases exponentially, the space cannot be covered in such a way that a global contraction of the posterior predictive variance is ensured, see e.g. (Kanagawa et al., 2018, Section 5.2) and (Wüthrich et al., 2021, Appendix C). Indeed, in the relatively small sample regime, there is at least one point $\boldsymbol{x}_n$ whose predictive variance is approximately equal to the (constant) prior variance, $\sigma_{n-1}^2(\boldsymbol{x}_n) \approx \sigma_0^2$. This is explained by the absence of neighboring observations, in which case the training data provides a negligible reduction of uncertainty because $\boldsymbol{c}_{n-1}(\boldsymbol{x}_n) \approx \boldsymbol{0}$. As a result, the worst-case instantaneous regret, being proportional to the predictive standard deviation (Lemma A.1), remains approximately constant for fixed $\beta_n$, or alternatively is nondecreasing for nondecreasing $\beta_n$ (Chowdhury & Gopalan, 2017, Theorem 2), and thus there is no guarantee of improvement.

These problems can be alleviated by more informative GP priors. In particular, spatially-varying prior variances allow the surrogate to remain spatially informative, even in the absence of neighboring observations, and spatially-varying lengthscales can encourage greedier strategies, in an attempt to improve upon the incumbent solution when given small budgets.

## 4 Informative Covariance Functions

For the construction of our informative covariance functions, we start with a set of *anchors* $\{\boldsymbol{x}_0^{(l)}\}$, i.e., a set of promising candidate solutions. This prior belief can be interpreted as a point-mass distribution over promising points. A more conservative prior $p(\boldsymbol{x}^\star)$ is obtained by adding an uninformative slab that ensures the optimal solution is included in its support, and forming a kernel density estimate by associating a non-negative weight $w_l$, distance function $d_l$ and kernel $k_l$ with each anchor $\boldsymbol{x}_0^{(l)}$,

$$p(\boldsymbol{x}^\star) \propto \phi(\boldsymbol{x}^\star), \qquad \phi(\boldsymbol{x}^\star) = 1 + \frac{1}{L}\sum_{l \leq L}(w_l - 1)\, k_l\left(d_l(\boldsymbol{x}^\star, \boldsymbol{x}_0^{(l)})\right). \tag{11}$$

Distance functions and kernels characterize the neighborhood of anchors. Larger weights correspond to the belief that better candidates can be found in a given neighborhood. Next, we introduce spatially-varying

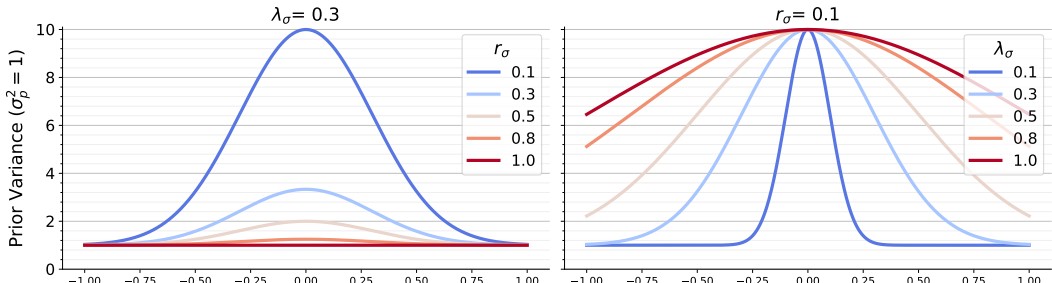

Figure 4: For a Gaussian $k_\sigma$ and weighted Euclidean distance $d_\sigma = d_{\mathrm{WE}}$, decreasing the ratio $r_\sigma$ increases the prior variance around the anchor $x_0 = 0$. Stationarity is recovered when $r_\sigma = 1$. The lengthscale $\lambda_\sigma$ in $d_{\mathrm{WE}}$ controls the rate at which the prior variance decreases to the stationary baseline.

prior variances by rescaling the stationary covariance function $C_{\mathrm{S}}(\boldsymbol{x}_i, \boldsymbol{x}_j)$,

$$C_{\mathrm{I}}(\boldsymbol{x}_i, \boldsymbol{x}_j) = \sigma_0^2(\boldsymbol{x}_i, \boldsymbol{x}_j)C_{\mathrm{S}}(\boldsymbol{x}_i, \boldsymbol{x}_j), \qquad \sigma_0^2(\boldsymbol{x}_i, \boldsymbol{x}_j) = \sigma_p^2\sqrt{\phi(\boldsymbol{x}_i)}\sqrt{\phi(\boldsymbol{x}_j)}, \qquad (12)$$

where $\sigma_p$ is a scaling constant and $C_{\mathrm{S}}(\boldsymbol{x}, \boldsymbol{x}) = 1$. The modulating $\sigma_0^2(\boldsymbol{x}_i, \boldsymbol{x}_j)$ is symmetric and separable in $\boldsymbol{x}_i$ and $\boldsymbol{x}_j$, and hence a valid covariance function (Genton, 2001). The informative covariance $C_{\mathrm{I}}(\boldsymbol{x}_i, \boldsymbol{x}_j)$, being the product of two covariance functions, is also a valid covariance function.

The *prior covariance function* $\sigma_0^2(\boldsymbol{x}_i, \boldsymbol{x}_j)$ is a generalization of the constant prior (co)variance $\sigma_0^2$. By recalling that covariance functions compute $\mathrm{Cov}(f(\boldsymbol{x}_i), f(\boldsymbol{x}_j))$, the intuition is as follows: For two points $\boldsymbol{x}_i$ and $\boldsymbol{x}_j$ that are close to anchors, i.e., large $\phi(\boldsymbol{x}_i)$ and $\phi(\boldsymbol{x}_j)$, this prior posits that their values, $f(\boldsymbol{x}_i)$ and $f(\boldsymbol{x}_j)$, are highly correlated because both should be close to the minimal value and hence similarly small. Then, if the probability for a point, e.g. $\boldsymbol{x}_j$, to be a good candidate decreases, the covariance becomes smaller because $f(\boldsymbol{x}_j)$, being less constrained, can take on a greater range of values.

In Section 5, we focus on the special case of a single anchor, for which we obtain

$$\sigma_0^2(\boldsymbol{x}_i, \boldsymbol{x}_j) = \sigma_p^2\sqrt{1 + \left(\frac{1}{r_\sigma} - 1\right)k_\sigma\left(d_\sigma(\boldsymbol{x}_i, \boldsymbol{x}_0)\right)}\sqrt{1 + \left(\frac{1}{r_\sigma} - 1\right)k_\sigma\left(d_\sigma(\boldsymbol{x}_j, \boldsymbol{x}_0)\right)}, \qquad (13)$$

where $k_\sigma \triangleq k_1$, $d_\sigma \triangleq d_1$ and $r_\sigma \triangleq 1/w_1 \in (0, 1]$. We find this parametrization more convenient because inverse weights are bounded. The *ratio* $r_\sigma$ controls the degree of stationarity, with $r_\sigma = 1$ resulting in a stationary covariance and $r_\sigma \to 0$ in increased nonstationary effects, as demonstrated in Figure 4.

Next, we introduce spatially-varying lengthscales via input warping. In input warping (Snoek et al., 2014; Risser, 2016), the stationary covariance function $C_{\mathrm{S}}$ is computed in a transformed space, i.e., after applying a (possibly nonlinear) warping function $h_\lambda$ to $\boldsymbol{x}$. Endowed with such a transformation, the informative covariance function $C_{\mathrm{I}}(\boldsymbol{x}_i, \boldsymbol{x}_j)$, in Equation 12, is given by

$$C_{\mathrm{I}}(\boldsymbol{x}_i, \boldsymbol{x}_j) = \sigma_0^2(\boldsymbol{x}_i, \boldsymbol{x}_j)C_{\mathrm{S}}\left(h_\lambda(\boldsymbol{x}_i), h_\lambda(\boldsymbol{x}_j)\right). \qquad (14)$$

Recall that inputs in $\sigma_0^2(\boldsymbol{x}_i, \boldsymbol{x}_j)$ are already being compared to anchors in the shaping function $\phi$ (Equation 11). Now, notice that an anisotropic stationary covariance function, equipped with the Mahalanobis distance $d_{\mathrm{M}}$, is equivalent to an isotropic covariance function, equipped with the standard Euclidean distance and linear transformation $h_\lambda(\boldsymbol{x}) = \boldsymbol{\Lambda}^{-\frac{1}{2}}\boldsymbol{x}$. Then, since the warping function can be any real vector-valued function, we propose a nonlinear transformation that shrinks the lengthscales $\lambda_d$ around anchors, thereby expanding their neighborhoods. In particular, for a single anchor, we have

$$h_\lambda(\boldsymbol{x}) = u_\lambda^{-\frac{1}{2}}(\boldsymbol{x})\boldsymbol{\Lambda}^{-\frac{1}{2}}\boldsymbol{x}, \qquad u_\lambda(\boldsymbol{x}) = 1 + (r_\lambda - 1)k_\lambda(d_\lambda(\boldsymbol{x}, \boldsymbol{x}_0)), \qquad (15)$$

with ratio $r_\lambda \in (0, 1]$.[2] Note that the matrix $\boldsymbol{\Lambda}$ is diagonal with squared lengthscales $\lambda_d^2$ and the neighborhood of $\boldsymbol{x}_0$ is characterized by the kernel $k_\lambda$ and distance $d_\lambda$. The shortest lengthscale is $\sqrt{r_\lambda}\lambda_d$.

---

[2] The general case with multiple anchors can be given by the kernel mixture $u_\lambda(\boldsymbol{x}) = 1 + 1/L\sum_{l \leq L}(1/w_l - 1)k_l(d_l(\boldsymbol{x}, \boldsymbol{x}_0^{(l)}))$, an expression similar to that of the shaping function $\phi$ in Equation 11. The inverse weights $1/w_l \in (0, 1]$ set shorter lengthscales.

## 5 Experiments

### 5.1 Setting

**Informative covariance functions** The shaping function $\phi$ should capture the existing knowledge about the objective, possibly by being elicited from experts or learned on problems that are known to be similar. This function, based on kernel mixtures, can express arbitrary beliefs, but its optimality crucially depends on its alignment with the shape of the objective function, as discussed in more detail in Section 3. For black-box optimization, in which little is known in advance, a default kernel and distance function may be used, and their hyperparameters learned by empirical Bayes. To avoid overfitting and reduce computational complexity, regularizing hyperpriors and hyperparameter tying are effective techniques that may also be used. In particular, the parameters and functions that control the spatially-varying prior (co)variance and lengthscales can be tied by setting the ratio $r_0 \triangleq r_\sigma = r_\lambda$, kernel $k_0 \triangleq k_\sigma = k_\lambda$ and distance $d_0 \triangleq d_\sigma = d_\lambda$. In our experiments, we specify a Gaussian $k_0$, equipped with a weighted Euclidean distance $d_0 = d_{\mathrm{WE}}$ and a shared lengthscale vector, which also parametrizes $\mathbf{\Lambda}$. Dirac delta hyperpriors for anchors have also been adopted, but note that none of these choices are strictly required. More implementation details can be found in Appendix B, and we refer to Appendix F for the settings of $r_0$, where we perform a sensitivity analysis.

An overview of the main methods using the proposed covariance functions (I) is included in Appendix D. These methods are I+XO and I+XA, where +XO denotes a fixed anchor $\boldsymbol{x}_0$ (placed at the center of the search space) and +XA denotes an adaptive anchor (placed at the incumbent solution). More methods can be developed by incorporating complementary methodologies. In this paper, we test I+XA+TR (trust-region optimization, Appendix D.1), I+XA+QM (informative mean functions, Appendix D.2), I+XA+SAAS (sparsity-inducing hyperpriors, Appendix H) and I+XA+GKEI (belief-augmented acquisition functions, Appendix I).

**Baselines** Previous work has assumed that optimal solutions are generally closer to the center of the search space rather than its boundary. As dimensionality increases, BO with a stationary covariance (S) may allocate a large portion of the budget toward blind exploration of the boundary, which by assumption does not contain the optimum, resulting in worse performance. To mitigate boundary overexploration, two approaches have been proposed: quadratic mean functions (+QM) (Snoek et al., 2015) and cylindrical covariance functions (C) (Oh et al., 2018). In particular, the latter are nonstationary covariance functions that aim to expand the innermost region, and have been shown to outperform stationary additive models on several high-dimensional additive objectives, without having to infer their structure. Further details can be found in Appendix C. Additionally, the use of trust regions (+TR), despite not specifically intended to address the boundary issue, has become a popular acquisition strategy for high-dimensional BO. This strategy can promote exploitation by constraining acquisitions to an adaptive region centered at the incumbent solution.

In our experiments, we refrain from direct comparisons with existing BO packages due to the diverse training and acquisition routines. Instead, we reimplemented the BO baselines to study the effect of different GP priors on optimization performance under controlled conditions. We also investigate the interplay between GP priors and trust regions. Our implementation of +TR follows TuRBO-1 (Eriksson et al., 2019). The GP models are global, trained on all observations, and the trust region is given by a box whose side lengths are doubled/halved after consecutive successes/failures. The hyperparameter values of +TR are identical to those of TuRBO-1, except for the failure tolerance, which is originally given by the number of dimensions. On a 100-dimensional problem, this means that the trust region is only shrunk after 100 consecutive failures, half of our evaluation budget. To increase exploitation, this tolerance is set to 10. Other baselines, such as the popular CMA-ES (Hansen, 2016), are described and shown in Appendix D.

**Performance metric** We report the normalized improvement over initial conditions (Hoffman et al., 2011),

$$\mathrm{NI}_n = \frac{f_{\mathrm{best}}^{(n_0)} - f(\boldsymbol{x}_{\mathrm{best}}^{(n_0+n)})}{f_{\mathrm{best}}^{(n_0)} - f(\boldsymbol{x}^\star)}, \qquad n = \{0, \dots, N\}, \tag{16}$$

where $n_0 = 16$ is the number of initial observations, $f_{\mathrm{best}}^{(n_0)}$ is the lowest initial value, $\boldsymbol{x}_{\mathrm{best}}^{(n_0+n)}$ is the incumbent solution, $\boldsymbol{x}^\star$ is the global minimizer and $N = 200$ is the number of acquisitions.[3]

---

[3]Experiments with a larger evaluation budget can be found in Appendix J.

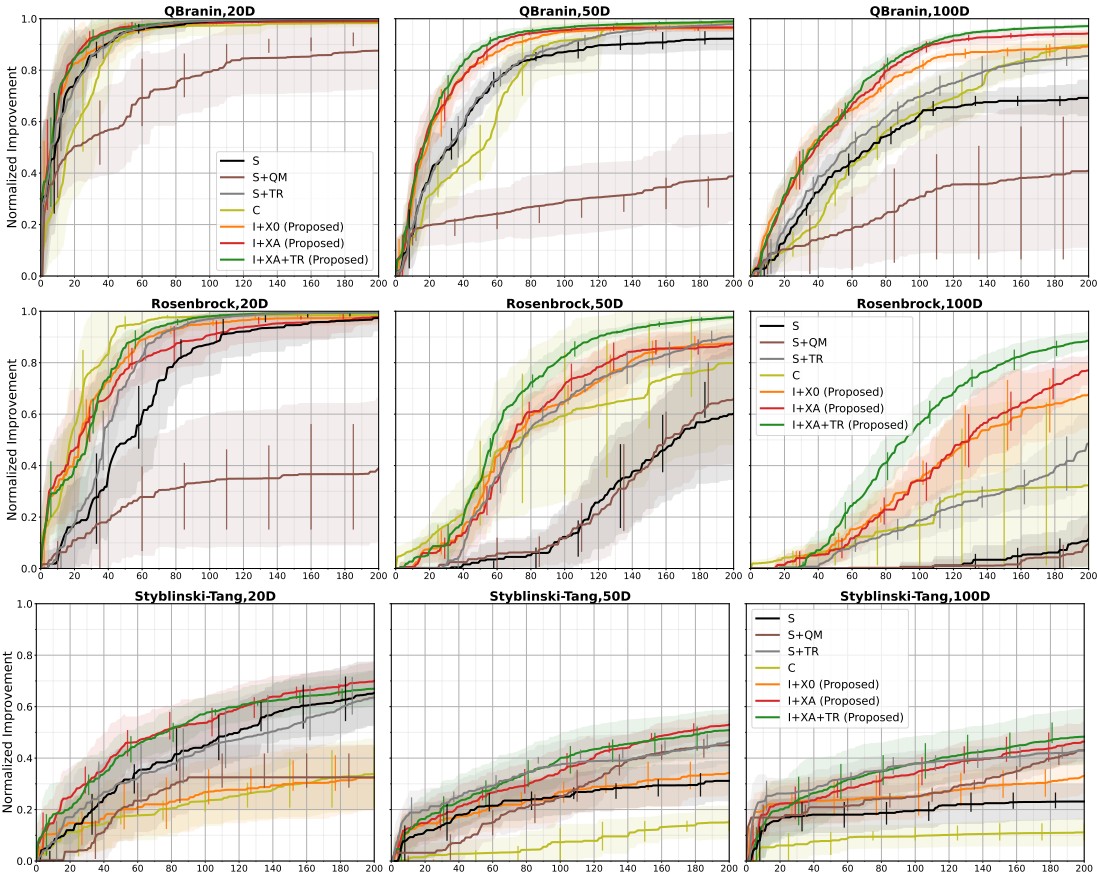

Figure 5: Performance on test functions with 20, 50 and 100 dimensions. Additional results are shown in Appendix D (methods and test functions) and Appendix J (larger evaluation budget). Solid curves and shaded regions represent the mean and standard deviation of the normalized improvement, computed over 10 trials with different initial conditions. Solid vertical lines indicate the interquartile range. Abbreviations: Stationary (`S`), Cylindrical (`C`) and Informative (`I`) covariances; Quadratic Mean (`+QM`); Origin (`+XO`) and Adaptive (`+XA`) anchors; acquisitions within Trust Region (`+TR`).

Unlike simple regret, this performance metric is invariant to initial conditions and to the range of $f$, always starting from 0 and only attaining the maximum value of 1 if a global optimum is found (zero simple regret). Additionally, our analysis does not hinge on the final best values because these can be deceiving. For instance, a method that rapidly achieves a good, though suboptimal, solution should not be considered inferior to another method that requires more evaluations, only to attain a marginally better value. The former is more sample efficient, which is a desirable feature when functions are expensive to evaluate. Some results are summarized by reporting the mean NI over $N$ acquisitions (or, equivalently, the normalized area under the NI curve), with 0 indicating worst and 1 optimal performance. Respective optimization trajectories are shown in Appendix D.

## 5.2 Results on Test Functions

We first focus on objectives whose landscapes are well-defined and can be easily controlled, allowing us to examine whether more informative priors translate into higher sample efficiency. Our test functions are roughly bowl-shaped, making the quadratic mean prior an appropriate choice for these problems. Test functions, such as Branin and Rosenbrock, have been previously used to test `C` (Oh et al., 2018). However, the Branin function is known for having multiple global optima, some of which are located close to the boundary. To overcome this, we design the QBranin function, which is obtained by adding a quadratic term, resulting in a bowl-shaped function with a single global optimum. The Styblinsky-Tang function, while multimodal, has a unique global optimum. A more detailed characterization of these test functions, as well as additional benchmarks (Levy and shifted objectives), can be found in Appendix B.3.

Table 1: Mean normalized improvement (over 200 acquisitions) on shifted (S35 and S50) Rosenbrock functions (mean ± standard deviation). Values > 0.5 suggest superlinear improvement rates. Additional methods, test functions and optimization trajectories can be found in Appendix D. Abbreviations: Stationary (`S`), Cylindrical (`C`) and Informative (`I`) covariances; Quadratic Mean (`+QM`); Origin (`+XO`) and Adaptive (`+XA`) anchors; acquisitions within Trust Region (`+TR`).

| | S35Rosenbrock | | S50Rosenbrock | |
| Method | 50D | 100D | 50D | 100D |
|---|---|---|---|---|
| `S` | $0.503 \pm 0.078$ | $0.283 \pm 0.052$ | $0.690 \pm 0.054$ | $0.411 \pm 0.048$ |
| `S+QM` | $0.507 \pm 0.086$ | $0.218 \pm 0.083$ | $0.548 \pm 0.193$ | $0.104 \pm 0.157$ |
| `S+TR` | $0.677 \pm 0.030$ | $0.486 \pm 0.030$ | $0.752 \pm 0.020$ | $0.535 \pm 0.061$ |
| `C` | $0.744 \pm 0.077$ | $0.487 \pm 0.067$ | $0.700 \pm 0.118$ | $0.377 \pm 0.142$ |
| `I+XO` | $\mathbf{0.803 \pm 0.031}$ | $\mathbf{0.644 \pm 0.026}$ | $\mathbf{0.831 \pm 0.038}$ | $\mathbf{0.729 \pm 0.027}$ |
| `I+XA` | $\mathbf{0.803 \pm 0.016}$ | $\mathbf{0.659 \pm 0.023}$ | $\mathbf{0.857 \pm 0.030}$ | $\mathbf{0.726 \pm 0.039}$ |
| `I+XA+TR` | $\mathbf{0.800 \pm 0.038}$ | $\mathbf{0.693 \pm 0.016}$ | $\mathbf{0.844 \pm 0.027}$ | $\mathbf{0.720 \pm 0.031}$ |

**Fixed anchor** The proposed covariance functions are initially tested with a fixed anchor placed at the center (`I+XO`). Figure 5 shows the performance curves on two objectives (top and middle) where the optimal solution is relatively close to this anchor. These results indicate that second-order nonstationary approaches, such as `C` and `I+XO`, can significantly outperform stationary methods, `S` and `S+QM`, particularly in higher-dimensional domains. While an informative mean (`S+QM`) is not necessarily better than an uninformative constant mean (`S`), prior information introduced via the covariance function leads to a consistent improvement over `S`. The reason may be that, in certain dimensions, `S+QM` has relatively large quadratic weights that hinder exploration away from the center, and in other dimensions, very small weights are unable to alleviate the boundary issue. Compared to `S`, `S+QM` and `C`, the proposed `I+XO` not only performs better on average but is also more reliable.

In Appendix E, an ablation study provides empirical evidence that incorporating both spatially-varying prior variances and lengthscales is important for tackling higher-dimensional problems. In particular, we there observe that the shorter lengthscales near anchors promote local exploration, which is especially effective when anchors are relatively close to optima. Meanwhile, spatially-varying prior variances can mitigate the boundary issue in high-dimensional domains, thereby improving sample efficiency. Next, we discuss the use of an adaptive anchor and the performance on the Styblinsky-Tang function.

**Adaptive anchor** The assumption that optimal solutions are close to the center is strong. As optima deviate from the center, the performance of `C` decreases markedly, eventually becoming comparable to that of `S` (Table 1). Despite the lower apparent vulnerability of `I+XO` to anchor misspecifications, this motivates the use of informative covariance functions with an adaptive anchor (`I+XA`). For simplicity, we opt for a greedy strategy where the anchor is given by the incumbent solution. For noise-free objectives, this is the point in the evidence set with the lowest observed value.

In addition to QBranin and Rosenbrock, Figure 5 includes the performance on the Styblinski-Tang function, whose optimal solution is relatively distant from the center. Notably, the center is a local maximum, being surrounded by exponentially many local modes. As the assumption that the global minimum is close to the center is no longer valid, we observe that the relative performance of `C` and `I+XO`, compared to `S` and `S+QM`, deteriorates. In particular, `C` becomes the worst-performing method and `I+XO` is worse than `S` in the 20-dimensional problem. Conversely, `I+XA` is consistently among the best-performing methods across all tests shown in the figure, and often outperforms the fixed-anchor variant (`I+XO`).

**Trust region** The use of an adaptive greedily-chosen anchor is partly motivated by trust-region optimization, wherein regions are centered at the incumbent solution. However, the proposed methodology is complementary to trust regions because it extends a stationary covariance function with spatially-varying parameters that are possibly aligned with prior beliefs about the objective. While `S+TR` can lead to significant improvements over `S`, overexploration of boundaries in a trust region can still occur because the model is stationary. The method `I+XA+TR` shows that the combination of informative covariance functions with trust-region optimization can improve sample efficiency.

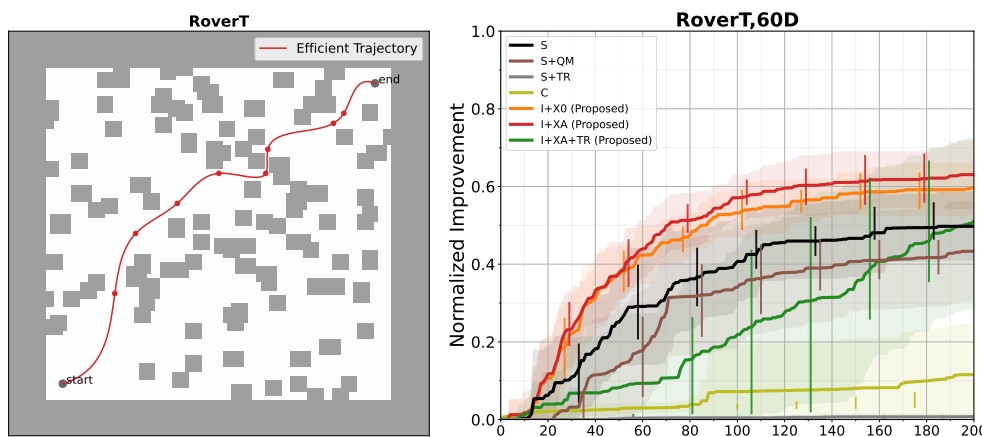

Figure 6: Map layout of the rover trajectory problem (left) and respective optimization results (right). Additional results and a more detailed analysis can be found in Appendix G. Solid curves and shaded regions represent the mean and standard deviation of the normalized improvement, computed over 10 trials with different initial conditions. Solid vertical lines indicate the interquartile range. Abbreviations: Stationary (`S`), Cylindrical (`C`) and Informative (`I`) covariances; Quadratic Mean (`+QM`); Origin (`+XO`) and Adaptive (`+XA`) anchors; acquisitions within Trust Region (`+TR`).

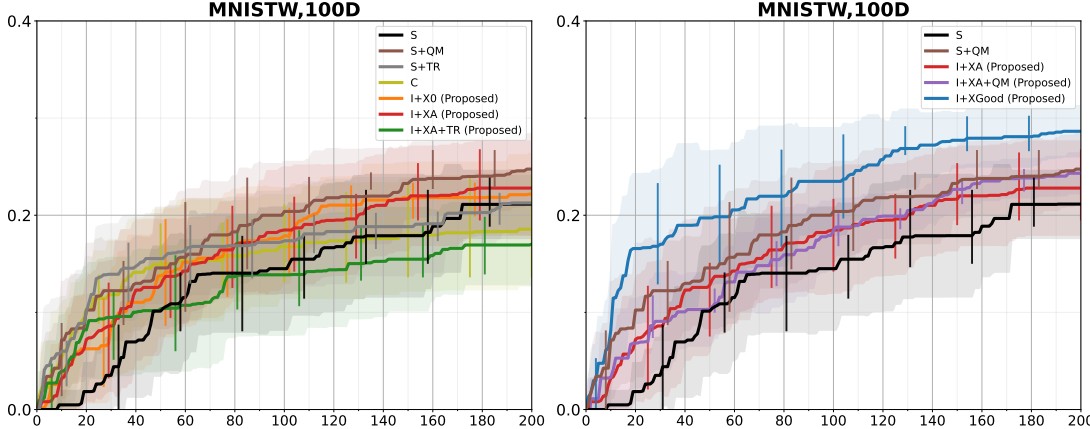

Figure 7: Bayesian optimization of 100 parameters of a neural network trained and tested on MNIST. Solid curves and shaded regions represent the mean and standard deviation of the normalized improvement, computed over 10 trials with different initial conditions. Solid vertical lines indicate the interquartile range. Abbreviations: Stationary (`S`), Cylindrical (`C`) and Informative (`I`) covariances; Quadratic Mean (`+QM`); Origin (`+XO`), Good (`+XGood`) and Adaptive (`+XA`) anchors; acquisitions within Trust Region (`+TR`).

### 5.3 Applications

**Rover trajectory planning** Recent studies have used the rover trajectory (RoverT) problem to showcase BO (Wang et al., 2018; Eriksson et al., 2019; Eriksson & Jankowiak, 2021). The goal is to optimize a trajectory such that the rover satisfies the target endpoints, $x_{\text{start}}$ and $x_{\text{end}}$, while avoiding collisions with objects, as depicted in Figure 6 (left). The trajectory is given by a B-spline that is fitted to 30 2-dimensional points, resulting in a 60-dimensional optimization problem. A more detailed description and analysis is provided in Appendix G. As we show in Figure 6 (right), `I+XO` and `I+XA`, which use an informative covariance, are the best-performing methods. We also observe that the performance of `S+TR` and `I+XA+TR` is significantly worse than that of `S` and `I+XA`, respectively. The reason may be that points in a trust region do not significantly improve upon the incumbent solution. For optimal performance, the endpoints must match the two targets because deviation incurs a significant loss. However, this may not be feasible when using a small box centered at a suboptimal solution, e.g. $x_{\text{best}} = 0$. Consecutive failures decrease the size of trust regions, potentially resulting in even slower progress.

**Neural network optimization** In this application, we optimize a neural network layer, as proposed by Oh et al. (2018). In particular, we perform BO of 100 parameters of a two-layer fully-connected network trained and tested on the MNIST dataset (LeCun, 1998). The architecture is $784 \xrightarrow{\mathbf{W}_1, \mathbf{b}_1} N_{\text{hidden}} \xrightarrow{\mathbf{W}_2, \mathbf{b}_2} 10$, with weights $\mathbf{W}$, biases $\mathbf{b}$, $N_{\text{hidden}} = 10$ and ReLU activations. The model is trained for 10 epochs with a batch size of 512 and evaluated on the test set using the negative log-likelihood loss. The weights $\mathbf{W}_2$ are found by BO using the test loss, while the remaining parameters are learned on the training set with Adam (Kingma & Ba, 2014). As pointed by Oh et al. (2018), the goal is not to find solutions that generalize well, but rather to evaluate the ability of BO to find solutions that perform well on the test set by minimizing the respective loss. The landscape is multimodal, and points at the boundary can yield relatively good solutions.

Results are shown in Figure 7. While S+QM appears to perform slightly better than I+XO and I+XA, models with an informative covariance function do not preclude the use of an informative mean function. For instance, we observe that I+XA+QM can improve upon I+XA toward the end, achieving similar average performance to that of S+QM. Relatedly, in Appendix D, we show that I+XA+QM outperforms both S+QM and I+XA on Styblinsky-Tang, demonstrating that informative mean and covariance functions are complementary. The best-performing method is I+XGood, which assumes strong prior information by using a fixed anchor placed at a solution found by S+QM. This highlights the value of high-quality information and that it can be incorporated into informative GP priors, without modifying the acquisition process.

### 5.4 Limitations

Despite the overall superior performance shown by methods with informative covariance functions, there is room for improvement that is left for future work. For instance, as dimensionality increases, the weighted Euclidean distance cannot differentiate points that are far only in a few dimensions. This can lead to difficulties in addressing the boundary issue in hypercubes. Potential remedies for this problem may include different parametrizations and distances based on the max-norm. Moreover, there is no mechanism in place that prevents greedily-chosen anchors from being close to the boundary. Since I+XA relies on empirical priors, it can be used without access to reliable prior information. However, as shown in Section 5.3 and Appendix D, methods that include reliable information can significantly outperform I+XA.

In Section 4, we provide formulas for covariance functions with multiple anchor points to demonstrate that the proposed methodology can in theory capture knowledge about the objective that goes beyond a single anchor. In our experiments, we focus on methods with a single anchor for several reasons. First, we wanted to show that higher sample efficiency is possible with simple design choices. Furthermore, baselines such as S+QM, S+TR and C implicitly assume the existence of a single anchor. By restricting ourselves to this case, the comparisons are fair, and we can assess the robustness of the different methods to anchor misspecification.

Additionally, in sequential myopic BO, which is the setup considered in this paper, it suffices to focus on a region that includes an optimum. The correct identification of such a region may be difficult in multimodal objectives, but as we have shown, the anchor can be dynamically adjusted. One option, without having to specify simultaneous anchors, is to maintain posterior beliefs about promising points and sample them at each step. If the anchors are not known, a similar strategy to that proposed by Eriksson et al. (2019) to maintain multiple trust regions is also a possibility. Other options include setting the anchors after performing cluster analysis and learning them by empirical Bayes. In this context, a different set of baselines would also be more appropriate, including methods with multiple trust regions and quadratic mean functions with multiple centers. Moreover, model selection can be used to automatically determine the number and location of anchors (see e.g. Appendix H). A careful treatment of informative covariance functions with multiple anchors requires substantial analysis and testing, and we believe this is best left for future work.

## 6 Related Work

**Second-order nonstationarity** Covariance functions with spatially-varying lengthscales were initially explored by Gibbs (1998) and Paciorek & Schervish (2006), and augmented with spatially-varying prior (and noise) variance by Heinonen et al. (2016) for GP regression. These models are derived from basis function expansions or process convolution (Risser, 2016), wherein the covariance function is obtained via integral transform of a base stochastic process with a kernel function, which may also depend on auxiliary mixing

variables (e.g. Wang et al., 2020a). In contrast, we introduce spatially-varying lengthscales via input warping, resulting in a simpler expression that does not require extra factors to ensure positive definiteness. Additionally, these other models place GP priors on the spatially-varying parameters and infer the respective latent functions by MCMC, being computationally expensive for high-dimensional parameter spaces. Instead, we use pointwise specifications of nonstationary effects, governed by semiparametric functions whose shape and parameters can be elicited from an expert or learned by empirical Bayes. Importantly, while these works focus on the estimation of the covariance function for regression purposes, we leverage second-order nonstationarity to incorporate information about promising points for optimization.

In the context of optimization, input warping was first studied by Snoek et al. (2014), focusing on monotonic input deformations via CDFs, specifically the Beta distribution. The motivation follows from a regression perspective, as the resulting GP is capable of approximating more realistic objective functions. More recently, Oh et al. (2018) proposed cylindrical covariance functions for optimization, outperforming the method by Snoek et al. (2014). In this case, after applying the cylindrical transformation, the radius is warped by the Kumaraswamy distribution, further expanding the innermost region of the search space, which is assumed to contain the optimal solution. A different methodology was explored by Martinez-Cantin (2015), who used linear combinations of local and global stationary covariance functions with input-dependent weights to induce spatially-varying lengthscales (piecewise constant). While the motivation is that the local assignment of shorter lengthscales promotes local exploration, the local component is not guaranteed to learn shorter lengthscales because both are initialized with the same hyperpriors. None of these works provide a regret analysis or consider spatially-varying prior (co)variances in their models.

**High-dimensional Bayesian optimization** To tackle the challenges presented by high-dimensional spaces, previous research has relied on certain structural assumptions (Binois & Wycoff, 2022), namely low effective dimensionality (e.g. Letham et al., 2020; Eriksson & Jankowiak, 2021) and additivity (e.g. Kandasamy et al., 2015; Gardner et al., 2017; Rolland et al., 2018). Space-partitioning schemes and trust regions have also been proposed (e.g. McLeod et al., 2018; Eriksson et al., 2019; Wang et al., 2020b; Diouane et al., 2021). Notably, these methods employ stationary covariance functions. Our work complements these by introducing more informative priors (see e.g. Appendices D and H). Cylindrical covariance functions (Oh et al., 2018) can also be augmented with spatially-varying (co)variances and anchors can be defined in the transformed space.

**Priors over the optimum** There have been limited attempts to incorporate beliefs about optimal solutions into BO. Souza et al. (2021) combined predefined prior distributions with the tree-structured Parzen estimator approach by Bergstra et al. (2011), which estimates the input given the output, as opposed to the standard GP approach of modeling the output given the input. By design, the former is restricted to the Expected Improvement acquisition function. Ramachandran et al. (2020) explored input warping via predefined CDFs that encode prior beliefs. More recently, Hvarfner et al. (2022) proposed belief-augmented acquisition functions. As we show in Appendix I, these functions and informative GP priors are complementary.

## 7  Conclusion

This work proposes informative covariance functions for Bayesian optimization, leveraging nonstationarity to incorporate input-dependent information. Our methodology relies on priors with spatially-varying parameters to express a priori preferences for certain regions and to promote multiscale exploration. These priors can more efficiently describe a wider class of objective functions and result in improved worst-case performance.

Our experiments showed that the proposed covariance functions can significantly increase sample efficiency, even under weak prior information, challenging the prevalent use of stationary models for optimization. Additionally, we showed that our work complements existing methodologies, including models with informative mean functions, trust-region optimization and belief-augmented acquisition functions. Despite our focus on continuous search spaces, we believe that this work can be extended to other types, such as discrete and mixed spaces, by adopting appropriate priors. Our methodology can also accommodate multiple anchors and be combined with other methods, relying for instance on additivity and low effective dimensionality. We recognize that priors over the possible locations of the optimum can be defined in transformed, possibly lower-dimensional spaces, but these details are left for future research. Overall, we see this work as a contribution toward the wider adoption of scalable nonstationary and informative models for optimization.

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

## A  Instantaneous Regret Bounds

**Lemma A.1.** *Assume the interval $|f(\boldsymbol{x}) - m_n(\boldsymbol{x})| \leq \beta_n \sigma_n(\boldsymbol{x})$ holds, $\forall \boldsymbol{x} \in \mathbb{X}$ and that global optima of acquisition functions can be found. Then, for popular acquisition functions, namely LCB and EI, and any covariance function, possibly nonstationary, the worst-case instantaneous regret depends on the width of the interval, being proportional to the posterior predictive standard deviation, $r_{n+1} \propto \sigma_n(\boldsymbol{x}_{n+1})$, where $\boldsymbol{x}_{n+1}$ is the acquisition at step $n + 1$.*

*Proof.* Let $\boldsymbol{x}^\star$ be the minimizer of $f(\boldsymbol{x})$. Then, based on (Srinivas et al., 2010, Lemma 5.2), we have the following instantaneous regret bound for LCB,

$$r_{n+1} = f(\boldsymbol{x}_{n+1}) - f(\boldsymbol{x}^\star) \tag{17}$$

$$\leq f(\boldsymbol{x}_{n+1}) - m_n(\boldsymbol{x}^\star) + \beta_n \sigma_n(\boldsymbol{x}^\star) \tag{18}$$

$$= f(\boldsymbol{x}_{n+1}) - \mathrm{LCB}(\boldsymbol{x}^\star) \tag{19}$$

$$\leq f(\boldsymbol{x}_{n+1}) - \mathrm{LCB}(\boldsymbol{x}_{n+1}) \tag{20}$$

$$= f(\boldsymbol{x}_{n+1}) - m_n(\boldsymbol{x}_{n+1}) + \beta_n \sigma_n(\boldsymbol{x}_{n+1}) \tag{21}$$

$$\leq 2\beta_n \sigma_n(\boldsymbol{x}_{n+1}). \tag{22}$$

Regarding EI, we know $B_n \mathrm{I}_n(\boldsymbol{x}) \leq \mathrm{EI}(\boldsymbol{x}) \leq \mathrm{I}_n(\boldsymbol{x}) + (\beta_n + 1)\sigma_n(\boldsymbol{x})$, by (Wang & de Freitas, 2014, Lemma 9), with $B_n = \tau(-\beta_n)/\tau(\beta_n) \leq 1$, and $\tau(z) = z F_{\mathcal{N}}(z) + \mathcal{N}(z; 0, 1)$, because $\tau$ is nondecreasing, $\tau'(z) = F_{\mathcal{N}}(z) \in [0, 1]$. Also, recall that the improvement is given by $\mathrm{I}_n(\boldsymbol{x}) = \max(0, m_n^- - f(\boldsymbol{x}))$ with $m_n^- = \min_{\boldsymbol{x}} m_n(\boldsymbol{x})$. Now, we derive a simpler, tighter upper bound on EI,

$$\mathrm{EI}(\boldsymbol{x}) = \sigma_n(\boldsymbol{x})\tau(z(\boldsymbol{x})) \tag{23}$$

$$\leq \sigma_n(\boldsymbol{x})\tau(0) \tag{24}$$

$$= \frac{1}{\sqrt{2\pi}}\sigma_n(\boldsymbol{x}) < \sigma_n(\boldsymbol{x}), \tag{25}$$

where the first inequality follows from $z(\boldsymbol{x}) = (m_n^- - m_n(\boldsymbol{x}))/\sigma_n(\boldsymbol{x}) \leq 0$ and nondecreasing $\tau$. Let $\boldsymbol{x}_n^- = \arg\min_{\boldsymbol{x}} m_n(\boldsymbol{x})$. Then, the instantaneous regret bound can be written as

$$r_{n+1} = f(\boldsymbol{x}_{n+1}) - f(\boldsymbol{x}^\star) \tag{26}$$

$$= (f(\boldsymbol{x}_{n+1}) - m_n^-) + (m_n^- - f(\boldsymbol{x}^\star)) \tag{27}$$

$$\leq (f(\boldsymbol{x}_{n+1}) - m_n^-) + \mathrm{I}_n(\boldsymbol{x}^\star) \tag{28}$$

$$\leq (f(\boldsymbol{x}_{n+1}) - m_n^-) + \frac{1}{B_n}\mathrm{EI}(\boldsymbol{x}^\star) \tag{29}$$

$$\leq (f(\boldsymbol{x}_{n+1}) - m_n^-) + \frac{1}{B_n}\mathrm{EI}(\boldsymbol{x}_{n+1}) \tag{30}$$

$$\leq (f(\boldsymbol{x}_{n+1}) - m_n^-) + \frac{1}{B_n}\sigma_n(\boldsymbol{x}_{n+1}) \tag{31}$$

$$\leq (m_n(\boldsymbol{x}_{n+1}) - m_n^-) + \left(\beta_n + \frac{1}{B_n}\right)\sigma_n(\boldsymbol{x}_{n+1}) \tag{32}$$

$$\leq \left(D_n(\boldsymbol{x}_{n+1}) + \beta_n + \frac{1}{B_n}\right)\sigma_n(\boldsymbol{x}_{n+1}) \tag{33}$$

$$\leq \left(D_n^+ + \beta_n + \frac{1}{B_n}\right)\sigma_n(\boldsymbol{x}_{n+1}), \tag{34}$$

where, in Equation 32, the difference is bounded as $|m_n(\boldsymbol{x}_{n+1}) - m_n^-| \leq D_n(\boldsymbol{x}_{n+1})\sigma_n(\boldsymbol{x}_{n+1})$, with $D_n(\boldsymbol{x}_{n+1}) = \sqrt{2\log(\sigma_n(\boldsymbol{x}_{n+1})/\sigma_n(\boldsymbol{x}_n^-))}$, by (Wang & de Freitas, 2014, Lemma 10). From the same lemma, we also know that $\sigma_n(\boldsymbol{x}_n^-)$ can be lower bounded by a positive value, and that there is a positive constant upper bounding $\sigma_n(\boldsymbol{x}_{n+1})$ by construction. Thus, for any covariance function, possibly nonstationary, there exists a positive constant $D_n^+$ such that $D_n^+ \geq D_n(\boldsymbol{x}_{n+1})$, which yields the result. $\qquad\square$

## B  Implementation Details

### B.1  Gaussian Processes

We used GPyTorch (Gardner et al., 2018) to implement all GP models in Python, including the proposed informative (I) covariance functions, cylindrical (C) covariance functions (Oh et al., 2018) and axis-aligned quadratic mean (+QM) (Snoek et al., 2015). Unlike the version provided by GPyTorch, our implementation of C follows the original training and prediction routines, which account for the special treatment of the origin, as discussed in more detail in Appendix C.

In order to ensure fair comparisons, all models are optimized by L-BFGS-B (Zhu et al., 1997) with a maximum of 1000 iterations per new acquisition and other options set to default values. In addition, the stationary components $C_S$ are Matérn ($\nu = 5/2$), equipped with the weighted Euclidean distance (lengthscales $\lambda_d$). Positivity constraints are handled internally by GPyTorch using the softplus transform, except for C-specific hyperparameters, which use the log transform (Appendix C). General hyperparameters are given uninformative priors with bounds from (Oh et al., 2018). These include the prior variance $\sigma_0^2 \sim \mathcal{U}(e^{-12}, e^{20})$ and lengthscales $\lambda_d \sim \mathcal{U}(e^{-12}, 2\sqrt{D})$, $d \in \{1 \ldots D\}$, where the upper bound corresponds to the maximum Euclidean distance in the centered hypercube $\mathbb{X} = [-1, 1]^D$. Table 2 summarizes the default settings of I. For noise-free objectives, the noise hyperparameter is set to a fixed value $\sigma_y^2 = 10^{-3}$.

By default, models are characterized by an uninformative prior mean function with a uniformly-distributed constant $b \sim \mathcal{U}(y_{\min}, y_{\max})$, where the bounds correspond to the minimum and maximum observed values. Variants +QM use an axis-aligned quadratic mean function $m(\boldsymbol{x}) = b + \sum_d a_d([\boldsymbol{x}]_d - [\boldsymbol{c}_0]_d)^2$, as suggested by Snoek et al. (2015) and Swersky (2017, Section 4.4). The quadratic weights are independent, distributed as $a_d \sim \mathcal{HS}^+(2)$. The half-Horseshoe is a spike-and-slab hyperprior with a spike at 0 and a slab on the positive real line, only allowing convex quadratic functions. Quadratic mean functions with adaptive, greedily-chosen centers $\boldsymbol{c}_0 = \boldsymbol{x}_{\text{best}}$ were tested, but no significant or consistent improvement compared to the default center $\boldsymbol{c}_0 = \boldsymbol{0}$ was observed. Hence, results using the former (+QM+XA) are only shown in Appendix D.

Table 2: Default settings of informative covariance functions.

| Quantity | Setting |
| --- | --- |
| Stationary corr. $C_S$ | Matérn ($\nu = 5/2$, $\sigma_0^2 = 1$) |
| Prior variance $\sigma_p^2$ | $\mathcal{U}(e^{-12}, e^{20})$ |
| Lengthscales $\lambda_d$ | $\mathcal{U}(e^{-12}, 2\sqrt{D})$ |
| Distances $d_\sigma$, $d_\lambda$ | Tied $d_0$ |
| Distance $d_0$ | Weighted Euclidean $d_{\text{WE}}$ (shared $\lambda_d$) |
| Kernels $k_\sigma$, $k_\lambda$ | Tied $k_0$ |
| Kernel $k_0$ | Gaussian |
| Ratios $r_\sigma$, $r_\lambda$ | Tied $r_0$ |
| Ratio $r_0$ | Kumar(3.164, 1000), |
| | Sensitivity analysis (Appendix F) |

### B.2  Acquisition

Prior to acquiring new data, an initial evidence set is formed. The set $\mathcal{D}_{n_0}$ includes the origin and 15 other pseudo-uniformly distributed points, drawn according to a scrambled Sobol sequence (Owen, 1998; Joe & Kuo, 2008). This set provides the initial conditions from which initial models are estimated and their performance measured. For noise-free objectives, the incumbent solution is the point with the lowest observed value $\boldsymbol{x}_{\text{best}}^{(n_0+n)} = \arg\min_{\boldsymbol{x} \in \mathcal{D}_{n_0+n}} f(\boldsymbol{x})$, $n_0 = 16$, $n = \{0, \ldots, N\}$, $N = 200$.

The default acquisition function is the Expected Improvement (EI), which is implemented in BoTorch (Balandat et al., 2020). Maximization is performed on the CPU via gradient-based optimization with multiple restarts. The 20 initial points for the optimization are selected from 20010 candidates, including 10 points

that are chosen heuristically and 20000 points that are randomly generated. The former are obtained by small Gaussian perturbations of the incumbent solution, and the latter are drawn from a scrambled Sobol sequence. The main difference between this procedure and the one implemented by Oh et al. (2018) is the use of L-BFGS-B via BoTorch, as opposed to Adam (Kingma & Ba, 2014).

## B.3 Test Functions

Test functions are set up in a way similar to that used by Oh et al. (2018), with their domains adjusted to the centered hypercube $\mathbb{X} = [-1, 1]^D$ by applying a linear transformation to the inputs. Additionally, original objectives are shifted and scaled to satisfy $f(\boldsymbol{x}^\star) = 0$ and $f(\boldsymbol{0}) = 100$. Since the resulting functions are non-negative, models are then estimated on log-transformed data, after adding a very small offset. In general, this leads to improved performance because the distribution of function values becomes more Gaussian-like. Performance metrics are however computed without the log transform.

**QBranin** The Branin function is originally evaluated on $[-5, 10] \times [0, 15]$, and has several global optima. This function is modified by adding a quadratic component (last term), transforming it into a bowl-shaped function with only one global optimum,

$$f(\boldsymbol{x}) = \left([\boldsymbol{x}]_2 - \frac{5.1}{4\pi^2}[\boldsymbol{x}]_1^2 + \frac{5}{\pi}[\boldsymbol{x}]_1 - 6\right)^2 + 10\left(1 - \frac{1}{8\pi}\right)\cos([\boldsymbol{x}]_1^2) + 10 + 5[\boldsymbol{x}]_1^2. \tag{35}$$

Higher-dimensional versions are obtained by additive repetition.

**Shifted QBranin** The shifted variants SQBranin and SSQBranin are derived from QBranin by shifting the inputs in the original space by 2 and 3, respectively. Additional experiments with these variants are included in Appendix D. Figure 8 provides a 2D illustration of these functions in the transformed space $[-1, 1]^2$.

**Rosenbrock** This function is originally evaluated on the hypercube $[-5, 10]^D$ and is given by

$$f(\boldsymbol{x}) = \sum_{d=1}^{D-1} \left[100([\boldsymbol{x}]_{d+1} - [\boldsymbol{x}]_d^2)^2 + ([\boldsymbol{x}]_d - 1)^2\right]. \tag{36}$$

Similar to QBranin, the global optimum is close to the origin in the transformed space, $\boldsymbol{x}^\star = -0.2 \times \boldsymbol{1}$. However, as shown in Figure 9, the Rosenbrock is characterized by narrow banana-shaped valleys, making global optimization more challenging.

**Shifted Rosenbrock** S35Rosenbrock, S50Rosenbrock, and S65Rosenbrock are three variants of the Rosenbrock function, designed to test the robustness of methods that assume good values near the origin. These functions are obtained by shifting the inputs so that the global minima in the transformed space are at $\boldsymbol{x}^\star = 0.35 \times \boldsymbol{1}$, $\boldsymbol{x}^\star = 0.50 \times \boldsymbol{1}$, and $\boldsymbol{x}^\star = 0.65 \times \boldsymbol{1}$. Figure 9 provides a 2D illustration of these functions.

**Levy** The original domain is $[-10, 10]^D$ and the function is defined as

$$f(\boldsymbol{x}) = \sin^2(\pi w_1) + \sum_{d=1}^{D-1}(w_d - 1)^2\left[1 + 10\sin^2(\pi w_d + 1)\right] + (w_i - 1)^2\left[1 + \sin^2(2\pi w_i)\right], \tag{37}$$

$$w_d = 1 + \frac{[\boldsymbol{x}]_d - 1}{4}, \qquad i \in \{1, \ldots, D\}. \tag{38}$$

As shown in Figure 10 (left), the Levy function is characterized by strong sinusoidal components. The global optimum is again close to the origin, specifically at $\boldsymbol{x}^\star = 0.1 \times \boldsymbol{1}$ in the transformed space.

**Styblinski-Tang** The original domain is $[-5, 5]^D$ and the function is given by

$$f(\boldsymbol{x}) = \frac{1}{2}\sum_{d=1}^{D}\left([\boldsymbol{x}]_d^4 - 16[\boldsymbol{x}]_d^2 + 5[\boldsymbol{x}]_d\right). \tag{39}$$

Joint optimization of the Styblinski-Tang function is difficult because it has exponentially many local modes $2^D - 1$. In contrast to QBranin, Rosenbrock and Levy, the global optimum is relatively far from the origin, $\boldsymbol{x}^\star \approx -0.58 \times \boldsymbol{1}$ in the transformed space. Figure 10 (right) provides an illustration of this multimodal function.

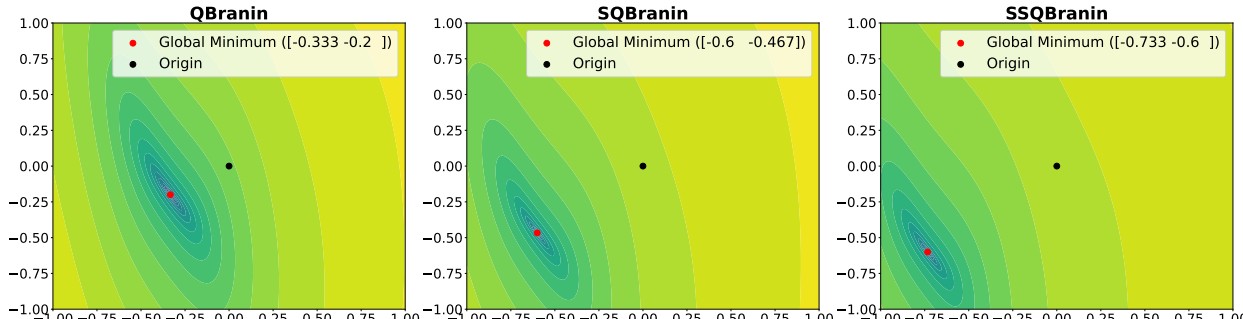

Figure 8: A 2D slice of QBranin and shifted variants. These test functions are unimodal and bowl-shaped. The purpose of SQBranin and SSQBranin is to test the robustness of methods that assume good values near the origin.

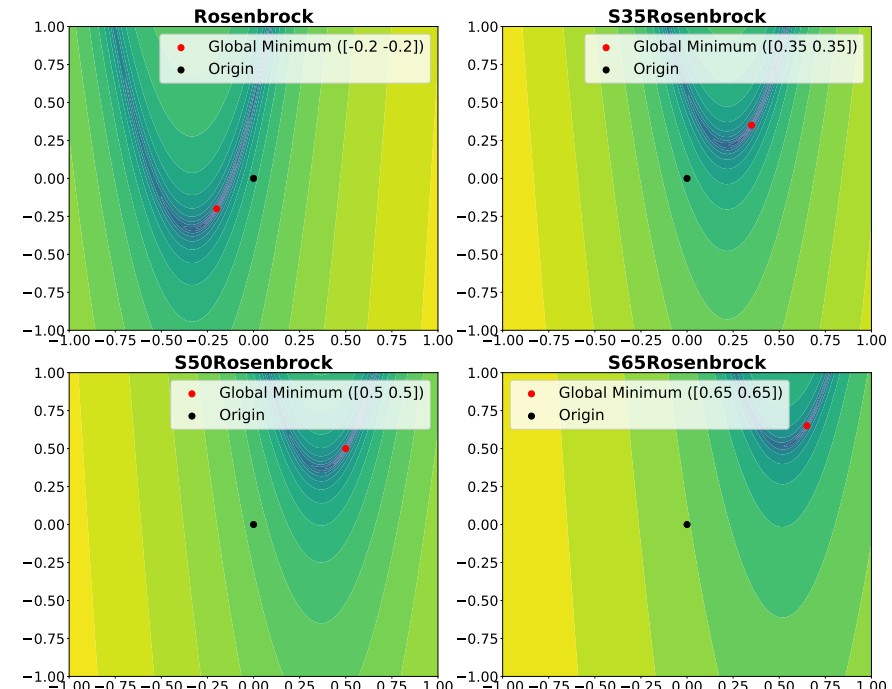

Figure 9: A 2D slice of Rosenbrock and three shifted variants (S35, S50 and S65).

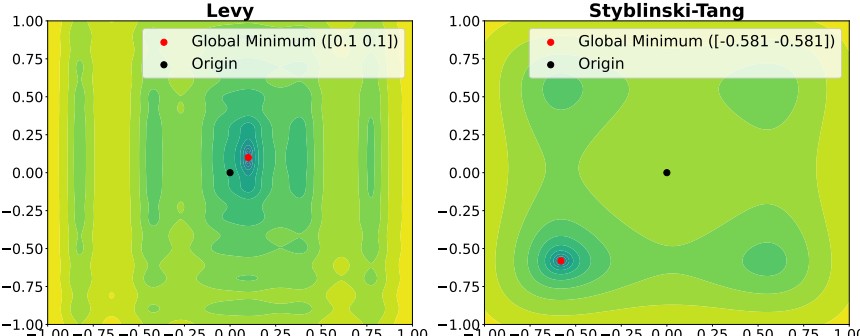

Figure 10: A 2D slice of Levy and Styblinski-Tang.

## C  Cylindrical Covariance Functions

Cylindrical covariance functions (Oh et al., 2018) aim to address the boundary issue that arises in high-dimensional Bayesian optimization. The cylindrical transformation $T(\boldsymbol{x})$ represents a point $\boldsymbol{x} \in \mathbb{X}$ in terms of its radius $r$ and angular components $\boldsymbol{a}$,

$$(r, \boldsymbol{a}) = T(\boldsymbol{x}) = \begin{cases} (\|\boldsymbol{x}\|_2, \boldsymbol{x}/\|\boldsymbol{x}\|_2), & \text{if} \quad \|\boldsymbol{x}\|_2 \neq 0 \\ (0, \boldsymbol{a}_{\text{arbitrary}}), & \text{if} \quad \|\boldsymbol{x}\|_2 = 0 \end{cases}. \tag{40}$$

where $\boldsymbol{a}_{\text{arbitrary}}$ is a random unit vector. Geometrically, this transformation maps balls of radius $R$ onto the surface of a cylinder of height $R$. A spherical shell of width $\delta r$ centered at a point $\boldsymbol{x}$ corresponds to a region in the Euclidean space whose volume increases exponentially with the radius $r$. As a result, algorithms that aim at equally covering each volume element in the Euclidean space spend exponentially more time at the boundary of the search space. In contrast, Oh et al. (2018) proposed to spend an equal amount of resources on each volume element in the transformed space. Intuitively, the transformation leads to a search in the Euclidean space that "expands the region near the center while contracting the regions near the boundary" (Oh et al., 2018, Section 3.2). The assumption is that optimal values are more likely to be found near the origin, so it is beneficial to encourage exploration in this region.

Remarkably, this transformation poses a challenge when comparing any non-origin point to the origin, since the latter is represented by an infinite set of points with radius 0. Oh et al. (2018) proposed using the point in the set that is closest to the point under comparison $\boldsymbol{x}$, setting $\boldsymbol{a}_{\text{arbitrary}} = \boldsymbol{x}/\|\boldsymbol{x}\|_2$. However, this solution has significant computational implications, requiring custom inference routines. If the origin is included in the training set, the Gram matrix must be recomputed for each test point. This issue is mitigated by block matrix inversion, where the block containing all non-origin points can be precomputed and reused.

Cylindrical covariance functions $C_{\text{cyl}}$ are defined by the product of a 1-dimensional Matérn ($\nu = 5/2$), measuring similarity of radii, and a function $C_a$ that compares angular components,

$$C_a(\boldsymbol{a}_1, \boldsymbol{a}_2) = \sum_{p=0}^{3} c_p (\boldsymbol{a}_1^\mathsf{T} \boldsymbol{a}_2)^p, \tag{41}$$

where $c_p$ is a normalized non-negative weight, given a log-Normal prior $\log(c_p) \sim \mathcal{N}(0, 2^2)$. To further encourage the expansion of the innermost region, Oh et al. (2018) used input warping on the radius component. This involves transforming the radius according to the Kumaraswamy cumulative distribution, $F_{\text{Kuma}}(r) = 1 - (1 - r^\alpha)^\beta$, with $\alpha \in [0.5, 1]$ and $\beta \in [1, 2]$. Both $\alpha$ and $\beta$ are given a spike-and-slab hyperprior on the log scale, with spike at $\log(1)$. Figure 11 shows possible warpings, including the identity transform obtained with $\alpha = 1$ and $\beta = 1$.

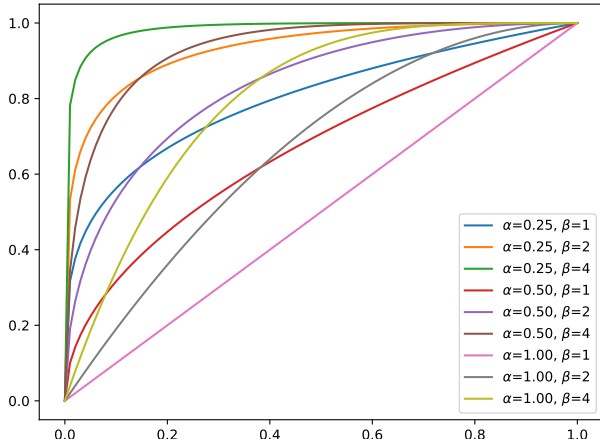

Figure 11: Input warping on the radius. Concave transformations expand regions of small radii.

# D  Additional Results

## D.1  Main Methods

**S** Bayesian Optimization (BO) with an uninformative Gaussian process (GP) prior, characterized by a constant mean and a stationary Matérn covariance function.

**S+QM** `S` with an axis-aligned quadratic mean function (Snoek et al., 2015; Swersky, 2017).

**S+TR** `S` with acquisitions within an adaptive trust region, i.e., a box centered at the incumbent solution that is shrunk/expanded based on consecutive failures/successes (Eriksson et al., 2019). For noise-free objectives, the incumbent solution is the point with the lowest observed value, as described in Appendix B.2.

**C** BO with a GP characterized by a constant mean and a cylindrical covariance function (Oh et al., 2018).

**I+X0 (Proposed)** BO with a GP featuring a constant mean and an informative covariance function. A single, fixed anchor is placed at the center (origin) of the search space.

**I+XA (Proposed)** `I+X0` with an adaptive greedily-chosen anchor, given by the incumbent solution. This is the same point as that used in `S+TR`.

**I+XA+TR (Proposed)** `I+XA` with a trust region, using the same adaptation scheme as `S+TR`.

## D.2  Additional Methods

**S+QM+XA** Quadratic mean function with a greedily-chosen center, given by the incumbent solution. In most tests, including the shifted objectives in Figures 12 and 14, `S+QM+XA` was outperformed by `S+QM`.

**GS+XA** Informative covariance functions are based on a search model for the optimum, which is given by a mixture distribution (Equation 11). For a single anchor, Gaussian kernel and weighted Euclidean distance, the equivalent distribution is a mixture of a uniform and a Gaussian distribution with diagonal covariance matrix. In `GS+XA`, the mean is set to the incumbent solution and the variance is estimated from the data. The standard deviation is the median absolute deviation (MAD) of the best 5% training points (rounded, with a minimum of 2), assuming that the coordinates are independently and identically distributed. The Gaussian mixing weight is set to 0.9. Other combinations involving a smaller mixing weight (0.5), a larger training set (10%) and a scale correction of the MAD estimate were also tested, but overall performed no better than `GS+XA`. In turn, `GS+XA` typically performed worse than `S` or `C`, as shown in Figures 12, 13 and 14.

**CMA-ES** Covariance Matrix Adaptation Evolution Strategy (Hansen, 2016) is a popular global optimization algorithm and is related to `GS+XA`, in that it samples from a Gaussian whose mean and covariance matrix are adaptive. In terms of implementation, we use pycma (Hansen et al., 2019) with a population size of 10. In general, `CMA-ES` was outperformed by `S`, as shown in Figures 12 and 14. On Styblinski-Tang (Figure 13), however, it performed well, but no better than `I+XA` or `I+XA+QM`.

**I+XA+QM (Proposed)** The uninformative constant mean in `I+XA` is replaced by the quadratic mean from `S+QM`. In tests where `S+QM` performed comparatively well, e.g. Styblinski-Tang (Figure 13), the combination `I+XA+QM` proved to be effective.

**I+XA+F (Proposed)** `I+XA` uses empirical priors, optimizing the hyperparameters via marginal likelihood. However, if additional information about the objective is available, better performance can be achieved. For instance, in cases where the center is already a good solution, e.g. Levy (Figure 13) and Rosenbrock (Figure 14), it can be difficult to improve upon initial conditions. In such cases, we can commit to a more exploitative search around `XA`. Based on this intuition, the focused method `I+XA+F` uses relatively short fixed lengthscales $\lambda_0 = 0.1\sqrt{D}$ to compute distances $d_0$, and a fixed $r_0 = 0.1$. The remaining hyperparameters are still learned from data. As future work, several priors can be used in tandem. A portfolio can then be managed using strategies similar to those proposed by Hoffman et al. (2011) and McLeod et al. (2018).

**I+XA+F+TR (Proposed)** In order to further encourage the search around `XA`, `I+XA+F` can also be combined with trust regions, achieving better performance in certain cases, e.g. Levy (Figure 13) and Rosenbrock (Figure 14).

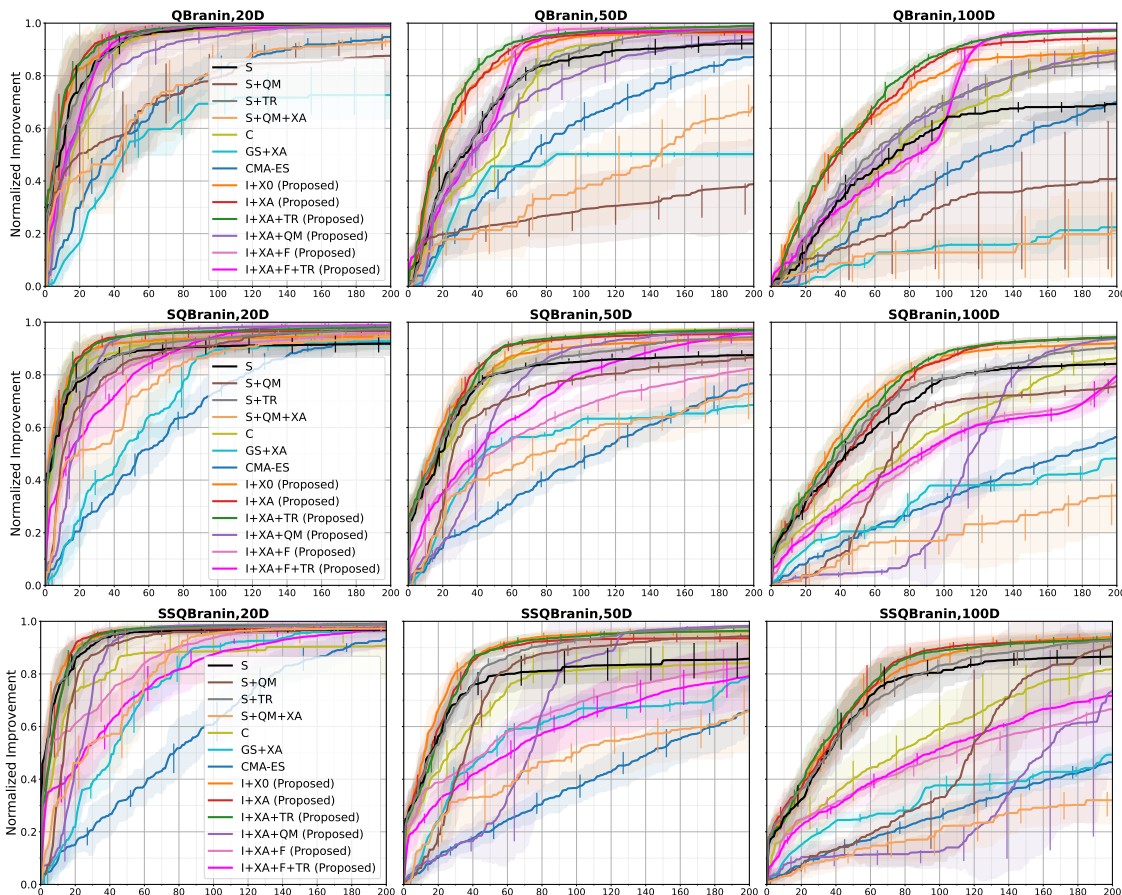

Figure 12: Performance on QBranin and variants, ranging from 20 to 100 dimensions. Solid curves and shaded regions represent the mean and standard deviation of the normalized improvement, computed over 10 trials with different initial conditions. Solid vertical lines indicate the interquartile range.

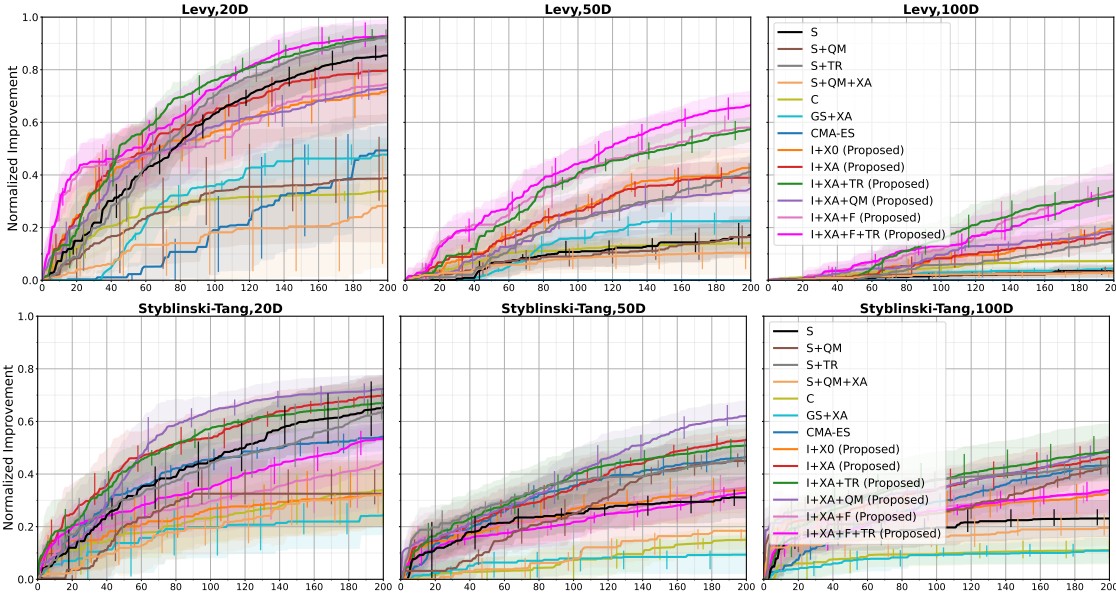

Figure 13: Performance on Levy and Styblinski-Tang, ranging from 20 to 100 dimensions. Solid curves and shaded regions represent the mean and standard deviation of the normalized improvement, computed over 10 trials with different initial conditions. Solid vertical lines indicate the interquartile range.

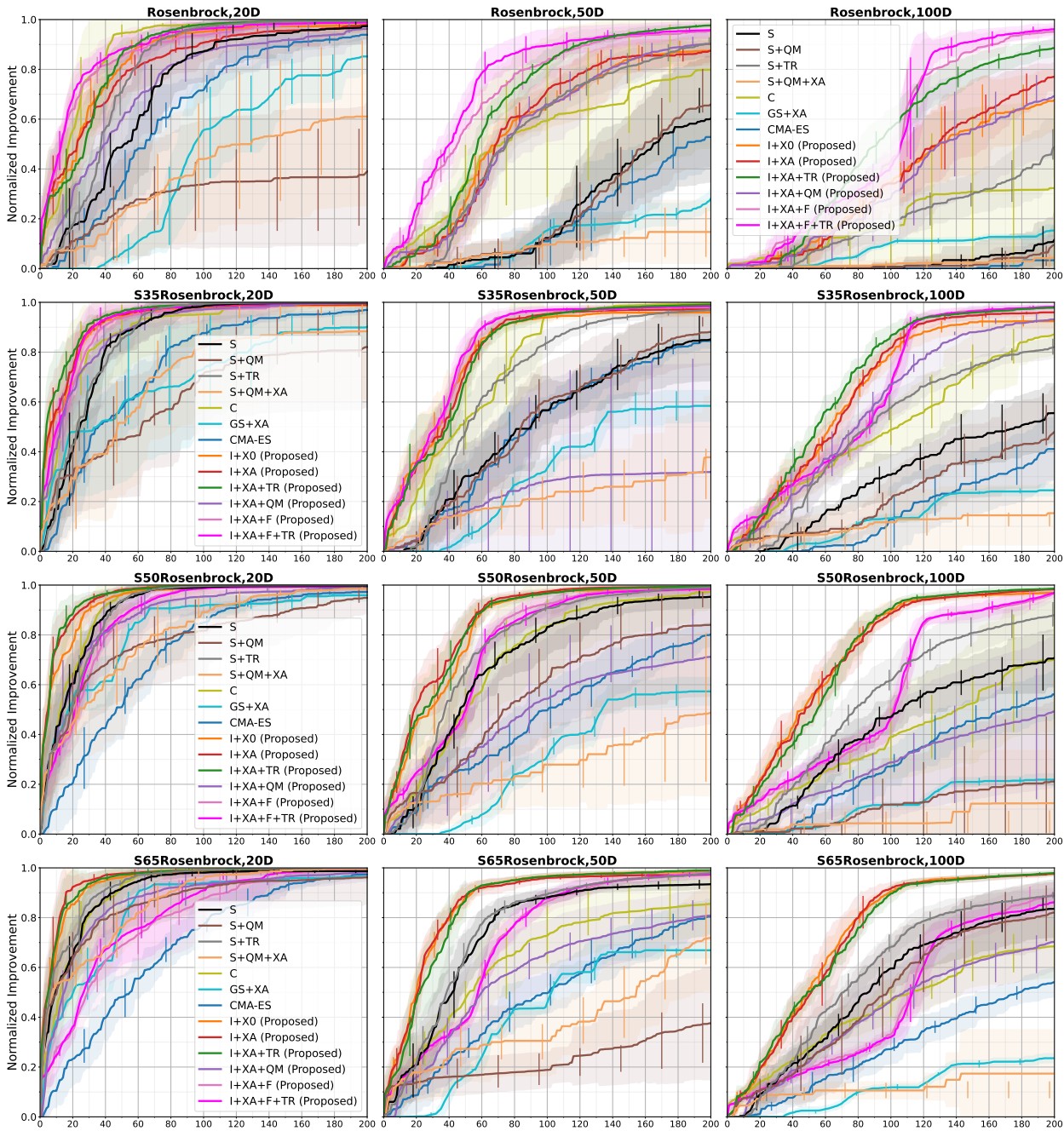

Figure 14: Performance on Rosenbrock objectives with 20, 50 and 100 dimensions. Solid curves and shaded regions represent the mean and standard deviation of the normalized improvement, computed over 10 trials with different initial conditions. Solid vertical lines indicate the interquartile range.

# E   Ablation Study

In this section, we investigate the contribution of spatially-varying prior variances (`V`) and lengthscales (`L`) toward more efficient high-dimensional optimization. For this purpose, we test these spatially-varying properties individually, referring to them as `I(V)` and `I(L)`.

Overall, the combination of spatially-varying prior variances and lengthscales is most effective in higher-dimensional spaces (50 and 100), where `I+X0` and `I+XA` generally outperform their `I(V)` and `I(L)` counterparts. In comparison to `S`, `I(V)` and `I(L)` prove to be effective on their own, but `I(L)` seems to account for most of the performance increase on Rosenbrock (Figure 15), where anchors are relatively close to the optimum. The shorter lengthscales around anchors promote local exploration, even after having acquired data in their neighborhood. Interestingly, when the anchor is misspecified (Styblinski-Tang), the performance of `I(L)+X0` is better than that of `I+X0` and similar to that of `S` in 20 dimensions. However, despite anchor misspecification, we observe the usefulness of spatially-varying prior variances as the dimensionality increases to 100, likely because these mitigate the boundary issue. For adaptive anchors, it is less clear whether the performance of `I+XA` on Styblinski-Tang is mostly due to spatially-varying prior variances or lengthscales. Both seem to be important because `I+XA` consistently outperforms `I(V)+XA` and `I(L)+XA`.

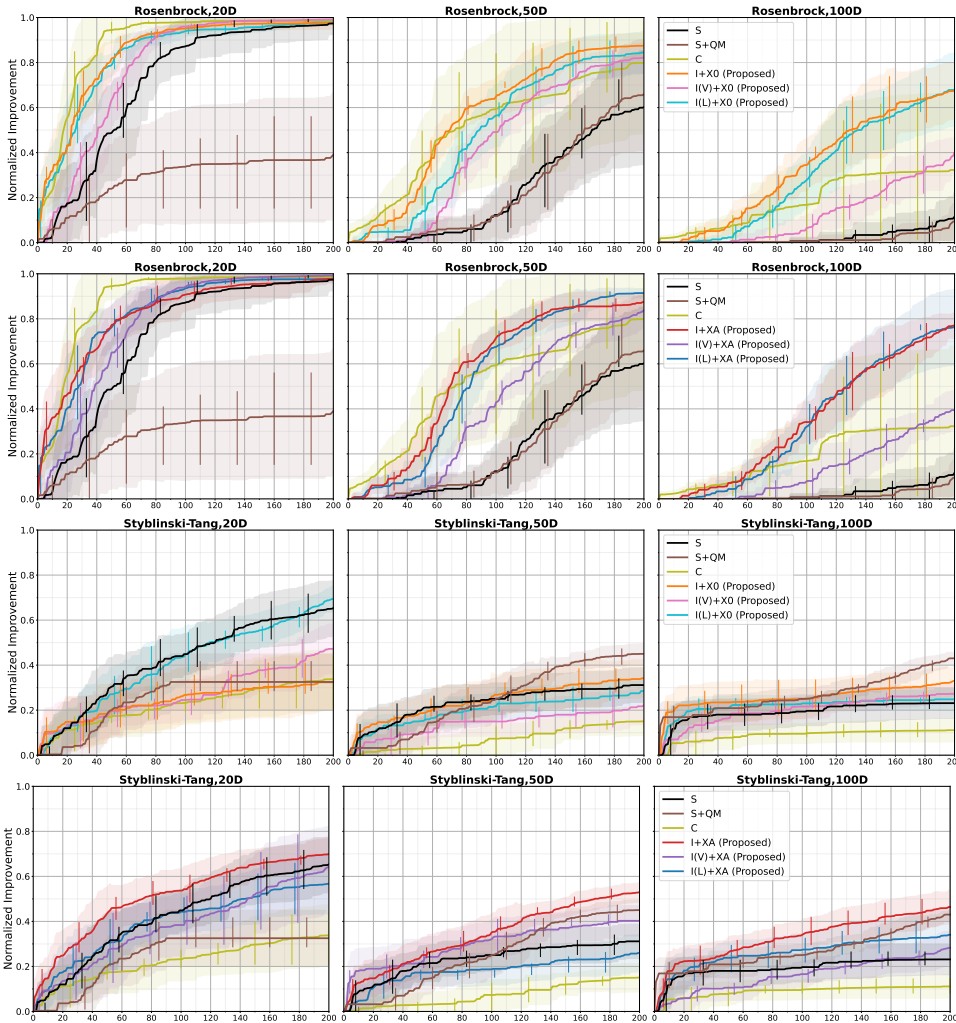

Figure 15: Ablation study. Solid curves and shaded regions represent the mean and standard deviation of the normalized improvement, computed over 10 trials with different initial conditions. Solid vertical lines indicate the interquartile range. Abbreviations: Stationary (`S`), Cylindrical (`C`) and Informative (`I`) covariances; Quadratic Mean (`+QM`); Origin (`+X0`) and Adaptive, greedily-chosen (`+XA`) anchors; spatially-varying prior Variances (`V`) and Lengthscales (`L`).

## F   Sensitivity Analysis

The informative covariance functions used in our experiments introduce a ratio $r_0 \in (0, 1]$, which leads to an uninformative stationary GP prior as it approaches 1. In terms of optimization, this hyperparameter balances global and local exploration under the informative search model (Equation 11) by adjusting the magnitude of spatially-varying prior (co)variances via $r_\sigma$ (Equation 13) and lengthscales through $r_\lambda$ (Equation 15). For example, compared to the stationary case, a ratio of $r_0 = 0.1$ indicates that the prior variance is up to $10\times$ larger and the squared lengthscales are $10\times$ shorter in the neighborhood of anchor $\boldsymbol{x}_0$.

By default, the ratio $r_0$ is learned by empirical Bayes and its prior is an informative Kumaraswamy density function with a peak at 0.1, given by Kumar$(3.164, 1000)$ and referred to as `K3`.[4] The Kumaraswamy distribution is related to the beta distribution and is similarly parametrized by two positive shape parameters, $a$ and $b$, with probability density function

$$p_{\text{Kumar}}(r; a, b) = abr^{a-1}(1 - r^a)^{b-1}, \qquad\qquad r \in (0, 1). \tag{42}$$

Notably, both $\text{Beta}(\cdot; \alpha, \beta)$ and $\text{Kumar}(\cdot; a, b)$ are special cases of the generalized $(p, \gamma, \delta)$-beta distribution (Jones, 2009, Equation 2.1), given by $(1, \alpha, \beta)$ and $(a, 1, b)$, respectively. However, the latter is more tractable because the beta function in the denominator simplifies as $\text{B}(1, b) = b^{-1}$. For a list of similarities and other advantages over the beta distribution, see (Jones, 2009, Section 7).

In the remainder of this section, we examine the performance of `I+XA` under different priors for the hyperparameter $r_0$. We tested uniform (`U`) and Dirac delta (`D`) priors, as well as several Kumaraswamy priors, with parameters chosen to yield densities that become increasingly narrower, while maintaining a peak at 0.1. In particular, we tested Kumar$(1.467, 10)$ denoted by `K1`, Kumar$(2.253, 100)$ denoted by `K2` and Kumar$(3.164, 1000)$ denoted by `K3`. Among these, `K1` is the broadest prior, as shown in Figure 16. The use of priors that favor smaller values of $r_0$ is motivated by the belief that the informative search model is useful. In this case, strategies that promote local exploration under this search model are more likely to improve upon initial conditions than those using an uninformative, uniform search model (stationary GP prior).

Figure 17 shows the performance of `I+XA` variants on QBranin, Rosenbrock and Styblinski-Tang. We begin by observing that, on QBranin, `I+XA` is relatively robust to the choice of prior over $r_0$ because the performance of all variants is similar and significantly better than that of `S`, `S+QM` and `C`. However, the differences become more noticeable on Rosenbrock and Styblinski-Tang. The uniform prior leads to the worst-performing variant, but `I+XA+U` can still outperform the other baselines on Rosenbrock. Conversely, narrower priors, such as `K2` and `K3`, which promote exploration under the informative search model, lead to the best-performing methods. Interestingly, `I+XA` with a delta prior (`I+XA+D`) can outperform other variants on Rosenbrock. This result suggests that learning all parameters via marginal likelihood may not be the best approach. Alternative training objectives that also consider optimization performance may provide better results, but this study is left for future work.

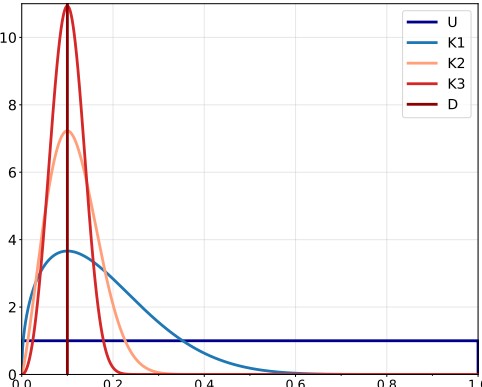

Figure 16: Probability density functions of the priors over $r_0$.

---

[4]The suffix `+K3` is omitted in other sections because it is the default prior over $r_0$. As discussed in Appendix D, the only exception is `I+XA+F` that uses $r_0 = 0.1$, corresponding to the Dirac delta (`D`) prior that is tested in this section.

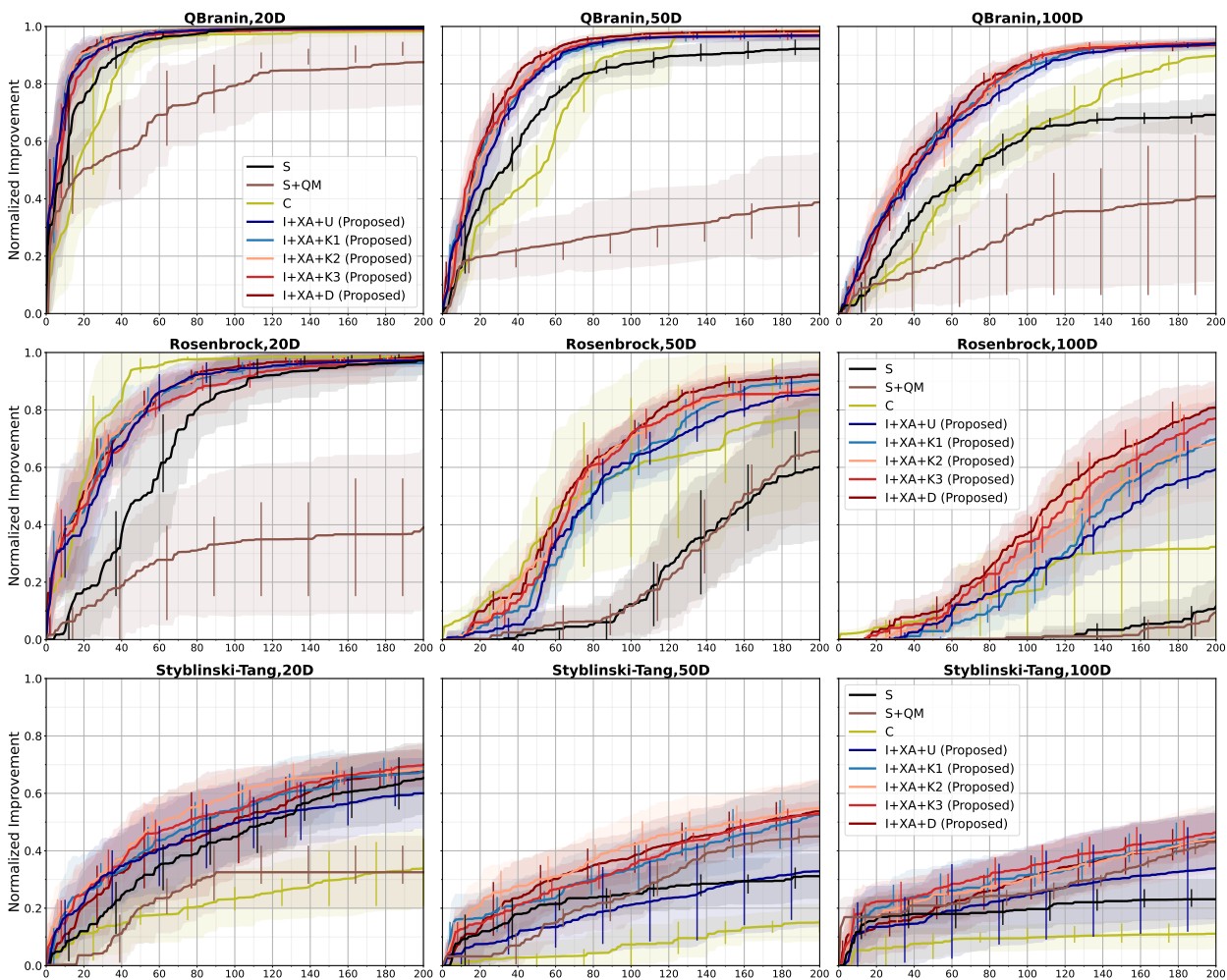

Figure 17: Performance of `I+XA` under different priors over $r_0$. Solid curves and shaded regions represent the mean and standard deviation of the normalized improvement, computed over 10 trials with different initial conditions. Solid vertical lines indicate the interquartile range. Abbreviations: Stationary (`S`), Cylindrical (`C`) and Informative (`I`) covariances; Quadratic Mean (`+QM`); Adaptive, greedily-chosen (`+XA`) anchors; Uniform (`+U`), Kumaraswamy (`+K`) and Dirac delta (`+D`) priors.

# G   Rover Trajectory Planning

In this section, we examine the rover trajectory problem proposed by Wang et al. (2018), which has also been adopted by more recent studies (Eriksson et al., 2019; Eriksson & Jankowiak, 2021). In brief, the goal is to optimize a 2D trajectory such that the rover starts as close as possible to $\boldsymbol{x}_{\text{start}}$ and stops near $\boldsymbol{x}_{\text{end}}$, while avoiding collisions with objects. The trajectory is given by a B-spline that is fitted to 30 2-dimensional points, resulting in a 60D optimization problem. The original reward function is defined as

$$f(\boldsymbol{x}) = c(\boldsymbol{x}) + \gamma \left( \|[\boldsymbol{x}]_{1,2} - \boldsymbol{x}_{\text{start}}\|_1 + \|[\boldsymbol{x}]_{59,60} - \boldsymbol{x}_{\text{end}}\|_1 \right) + b, \qquad \boldsymbol{x} \in [0,1]^{60}, \qquad (43)$$

where $\gamma = 10$, $b = 5$ and $c(\boldsymbol{x})$ is a nonpositive function that penalizes collisions. For minimization, we turn the reward into a loss function. In particular, we take the negative reward and apply a $+b$ shift, yielding a non-negative objective to be minimized. Models are again estimated on log-transformed data. The domain is also adjusted to the centered hypercube $\mathbb{X} = [-1,1]^D$ by applying a linear transformation to the inputs.

Figure 18 shows the original map layout with two overlaid trajectories found by S and I+XA (left), as well as the corresponding performance curves (right). In this problem, we immediately observe that the trajectories do not pass through the 2D points. This outcome is due to oversmoothed splines, which explains the performance of S. Intuitively, given the map layout, we would expect points near the boundary to yield penalized trajectories, but this is only true if the trajectories pass through the specified points. Instead, points near the boundary have a significant effect on the oversmoothed trajectories. The overexploration of boundaries that is characteristic of S in higher-dimensional problems confers an advantage in this situation. However, we observe that I+XA is eventually able to find trajectories of similar cost.

In order to align the problem formulation with our expectations, we modify the procedure that fits B-splines to the collection of 2D points. In terms of implementation, the function call that performs interpolation, `scipy.interpolate.splprep` (Virtanen et al., 2020), is extended with a smoothing factor s that is set to 0, which forces the trajectories to pass through all points. An example is shown in Figure 19 (left). Additionally, we clamp all trajectories into the admissible range and increase the cost for deviating from the endpoints, $\gamma = 50$. The latter promotes unfolded trajectories away from the initial incumbent solution $\boldsymbol{x}_{\text{best}}^{(n_0)} = \boldsymbol{0}$, as the relative cost of collisions is too steep otherwise. Performance curves for all rover tests are shown in Figure 20. Notably, we observe that S is no longer among the best-performing methods because the specification of points near the boundary correctly yields penalized trajectories.

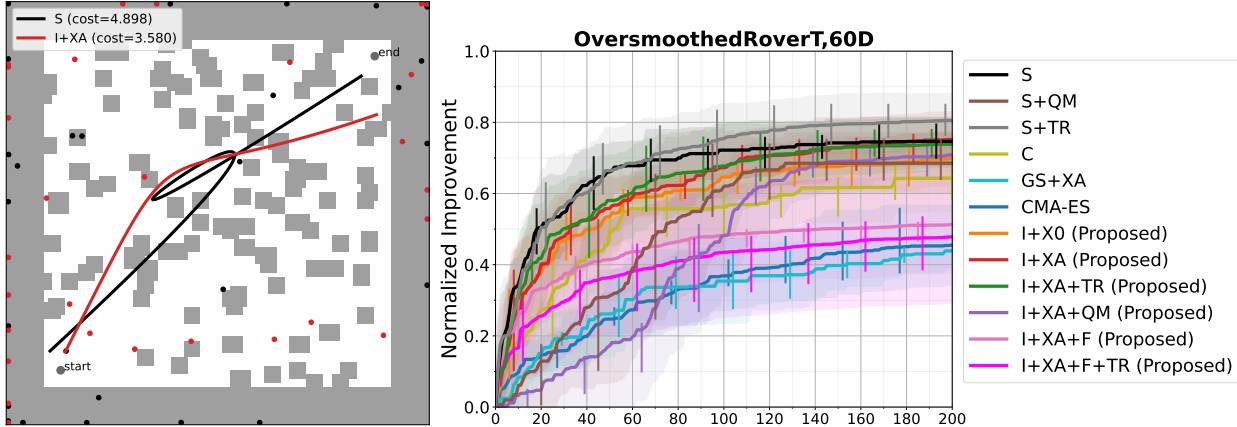

Figure 18: Original map layout with overlaid rover trajectories (left). Trajectories are given by a B-spline that is fitted to the 30 2-dimensional points. Performance on the original rover trajectory problem (right). Solid curves and shaded regions represent the mean and standard deviation of the normalized improvement, computed over 10 trials with different initial conditions. Solid vertical lines indicate the interquartile range.

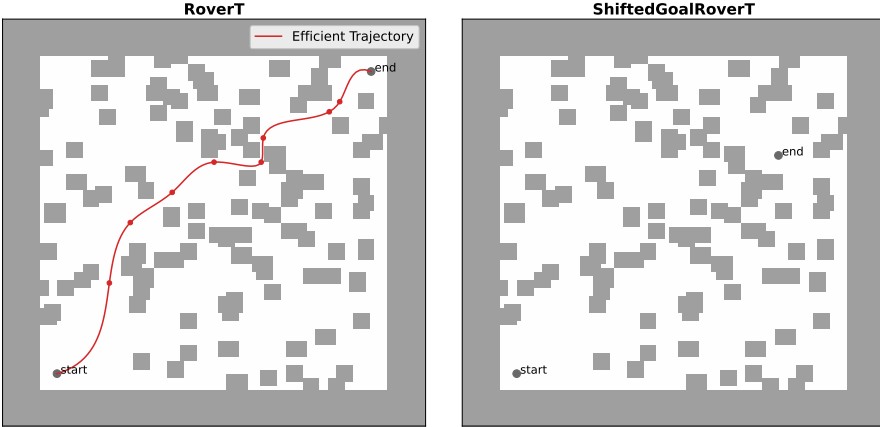

Figure 19: Original map layout with an overlaid rover trajectory (left). In RoverT, trajectories must pass through the specified collection of 2D points. The handcrafted trajectory demonstrates that it is not necessary to specify 30 different points. However, the reward function does not penalize less efficient trajectories, which leads to an infinite number of equally-good trajectories. As an additional test, we consider the same map layout with a shifted endpoint $\boldsymbol{x}_{end}$ (right).

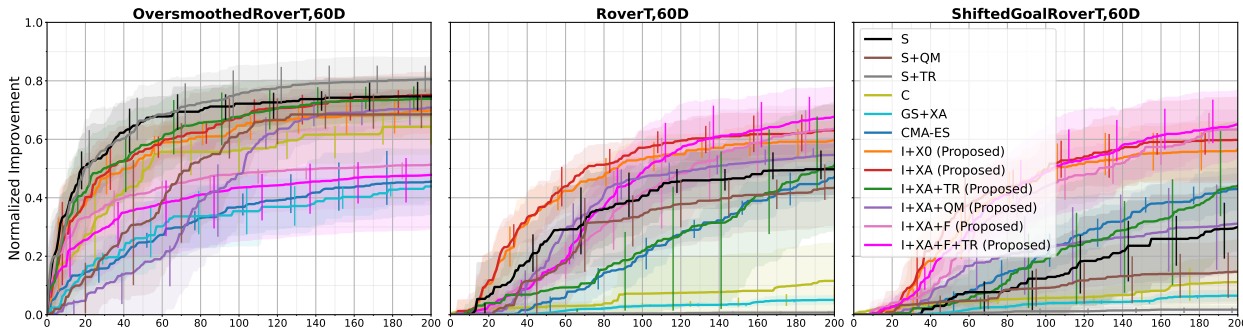

Figure 20: Performance on the rover trajectory problem. Solid curves and shaded regions represent the mean and standard deviation of the normalized improvement, computed over 10 trials with different initial conditions. Solid vertical lines indicate the interquartile range.

As shown in Figure 19 (left), an optimal trajectory does not require the specification of 30 different points. However, the reward function does not take into account the distance traveled, which means that the rover is free to roam the environment, provided that it starts at $\boldsymbol{x}_{\text{start}}$ and ends at $\boldsymbol{x}_{\text{end}}$ without colliding with other objects. For this reason, trajectories of similar cost can be significantly different and the choice of GP priors becomes even more important.

The importance of GP priors is further demonstrated in Figure 21, which shows an example of the best RoverT solutions found by BO. In particular, the stationary covariance in `S` (top-left) does not promote exploration around any particular region. In this case, the best trajectories tend to cover the entire space, while trying to avoid collisions. In contrast, the quadratic mean prior (top-middle) and the cylindrical covariance function (top-right) promote exploration near the center. As quadratic weights can be nearly zero, some points along the trajectory can also be near the boundary. In both cases, the prior dominates to such an extent that the endpoints of the best trajectories are comparatively far from the target endpoints. Notably, `I+X0` (bottom-left) also encourages exploration around the center, but to a lesser degree than `S+QM` and `C`, leading to better solutions. Finally, `I+XA` (bottom-middle) and `I+XA+F` (bottom-right) find the most cost-effective and efficient trajectories because both promote exploration around the incumbent solution, emphasizing the search for fine-tuned trajectories.

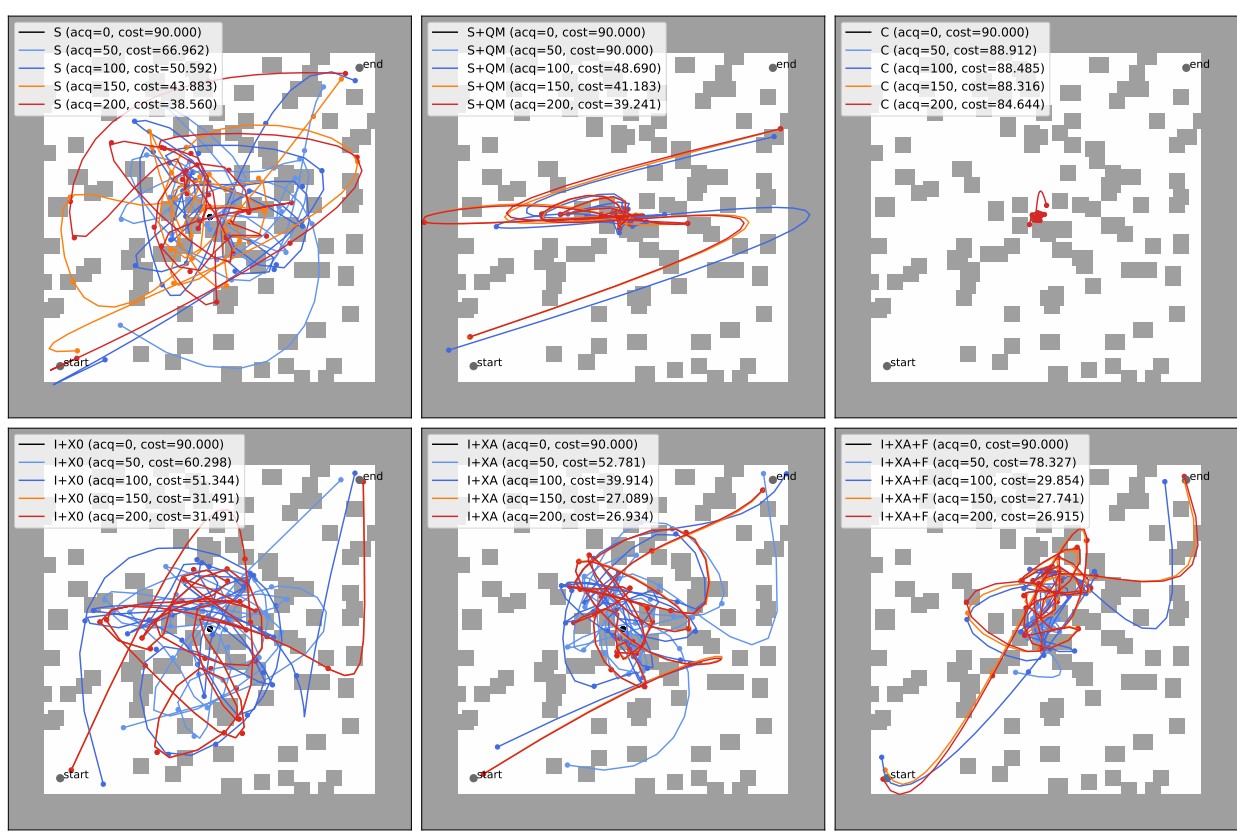

Figure 21: Example of the best rover trajectories found by BO using different GP priors. Top row: baselines S, S+QM and C. Bottom row: proposed methods I+X0, I+XA and I+XA+F.

# H    Sparse Axis-Aligned Subspaces (SAAS) Hyperprior

The Sparse Axis-Aligned Subspaces (`SAAS`) hyperprior, introduced by Eriksson & Jankowiak (2021), assumes that only a subset of dimensions affects the objective function. This sparsity-inducing hyperprior posits that the inverse squared lengthscales are distributed according to a half-Cauchy, $1/\lambda_d^2 \sim \mathcal{HC}(\tau)$, where $\tau$ is a global shrinkage hyperparameter. At each step, multiple GP models are trained by empirical Bayes with $\tau \in \{1, 10^{-1}, 10^{-2}, 10^{-3}\}$. The GP model with the highest leave-one-out cross-validation likelihood is then used for acquisition.

We evaluated a modified version of `S`, which incorporates `SAAS` and follows the original training procedure described above. As shown in Figures 22 and 23, `S+SAAS` generally performs worse than `S`, indicating that this hyperprior may adversely affect performance when the low effective dimensionality assumption does not hold. To further demonstrate the complementary nature of our proposed methodology, we tested `I+XA+SAAS`. This combination proved to be beneficial on QBranin, Rosenbrock and Levy. Recall that, in `I+XA`, the lengthscales also govern the nonstationary effects, and incorporating `SAAS` allows for longer lengthscales (c.f. Table 2). On the Styblinski-Tang function, however, this hyperprior continued to negatively impact performance, renewing the question of its suitability when the objective does not exhibit low effective dimensionality.

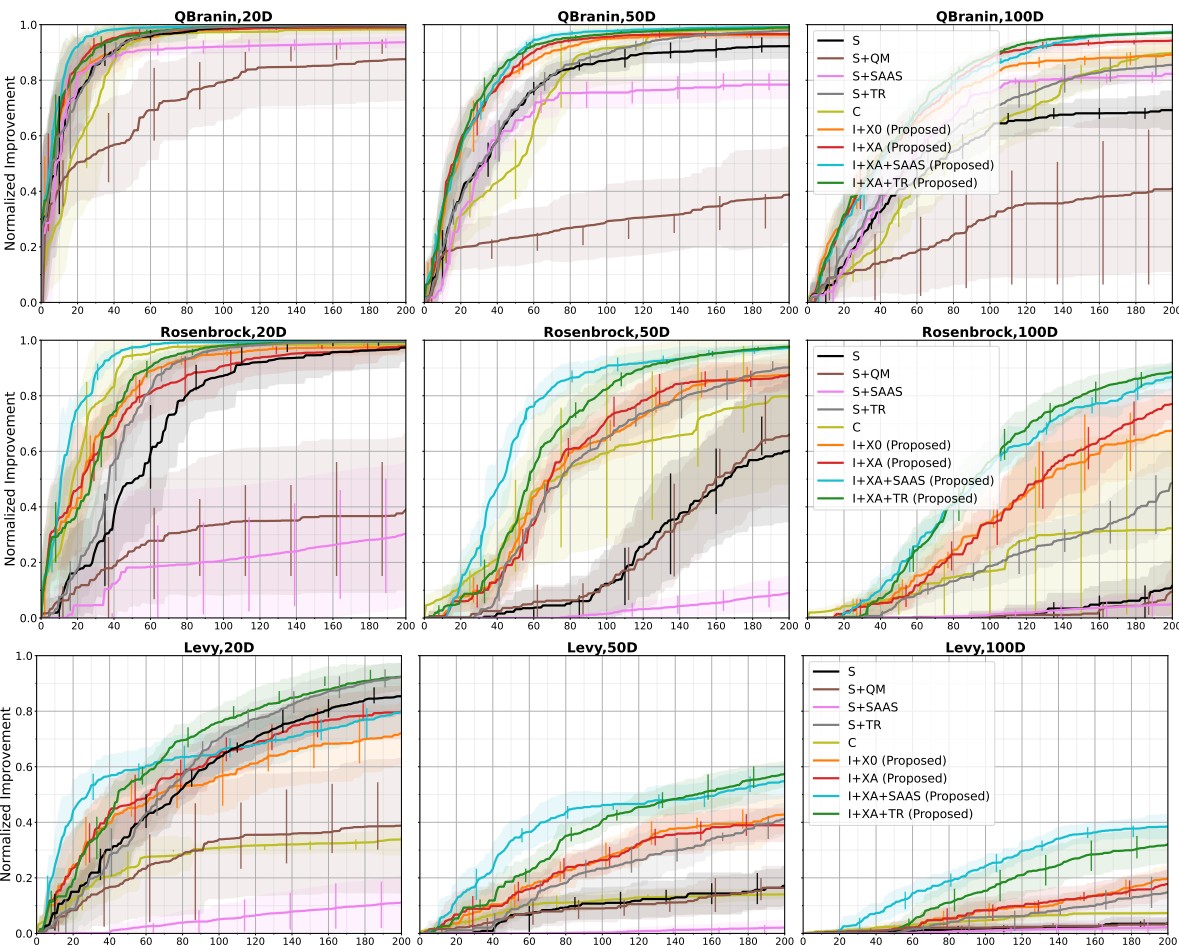

Figure 22: Performance on test functions, ranging from 20 to 100 dimensions. Solid curves and shaded regions represent the mean and standard deviation of the normalized improvement, computed over 10 trials with different initial conditions. Solid vertical lines indicate the interquartile range. Abbreviations: Stationary (`S`), Cylindrical (`C`) and Informative (`I`) covariances; Quadratic Mean (`+QM`); Origin (`+X0`) and Adaptive (`+XA`) anchors; acquisitions within Trust Region (`+TR`); Sparse Axis-Aligned Subspace (`+SAAS`) hyperprior.

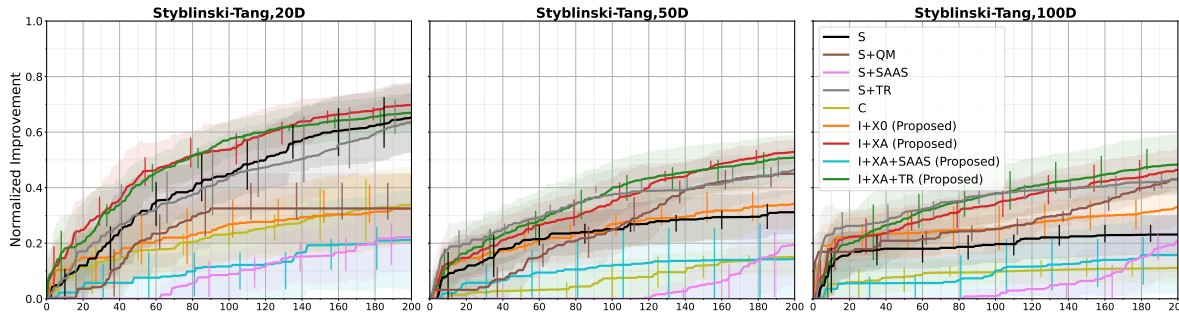

Figure 23: Performance on the Styblinski-Tang function with 20, 50 and 100 dimensions. Solid curves and shaded regions represent the mean and standard deviation of the normalized improvement, computed over 10 trials with different initial conditions. Solid vertical lines indicate the interquartile range. Abbreviations: Stationary (`S`), Cylindrical (`C`) and Informative (`I`) covariances; Quadratic Mean (`+QM`); Origin (`+X0`) and Adaptive (`+XA`) anchors; acquisitions within Trust Region (`+TR`); Sparse Axis-Aligned Subspace (`+SAAS`) hyperprior.

# I Belief-Augmented Acquisition Functions

In order to incorporate beliefs about optimal solutions into BO, Hvarfner et al. (2022) proposed prior-weighted acquisition functions, which are obtained by multiplying an acquisition function $\alpha$ with a prior distribution over possible locations of the optimum. As optimization progresses, the influence of the fixed prior $\pi(\boldsymbol{x})$ decays according to $\alpha_{\pi,n}(\boldsymbol{x}) \triangleq \alpha(\boldsymbol{x} \mid \mathcal{D}_n, \mathcal{M})\pi(\boldsymbol{x})^{\zeta/n}$, where the hyperparameter $\zeta$ controls the decay. As more acquisitions are made, the prior approaches a uniform distribution, only tilting the acquisition step initially, when the exponent $\zeta/n$ is not small. By default, Hvarfner et al. (2022) used Gaussian-Weighted EI (`GWEI`) and $\zeta = N/10$, where $N$ is the evaluation budget. The default Gaussian prior is characterized by a diagonal covariance matrix, with standard deviation set to 25% of the domain.

We evaluated two additional baselines, `S+GWEIX0` and `S+GWEIXA`, which use Gaussian-Weighted EI with fixed and adaptive locations. However, the results on high-dimensional test functions indicate that these baselines perform poorly when compared to `S`, as shown in Figures 24 and 25. We found that poor performance was also linked to machine precision because EI, whose values are already small, is multiplied by small factors, resulting in an acquisition function that is difficult to optimize in high dimensions. In order to improve performance, we removed the normalizing constant in the Gaussian prior, obtaining a Gaussian Kernel-weighted EI (`GKEI`). The method `I+XA+GKEI` shows that combining informative covariance functions and belief-augmented acquisition functions can be an effective strategy.

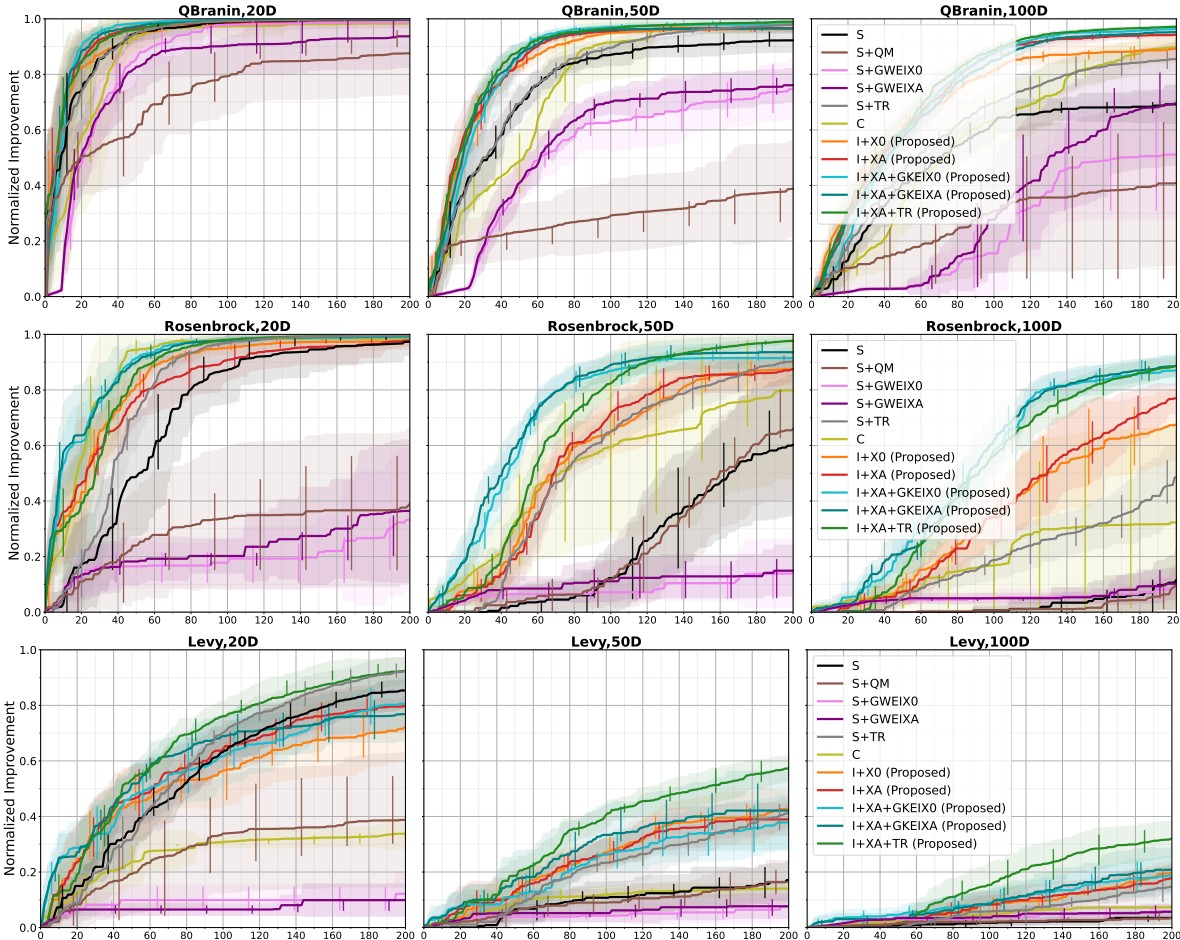

Figure 24: Performance on test functions, ranging from 20 to 100 dimensions. Solid curves and shaded regions represent the mean and standard deviation of the normalized improvement, computed over 10 trials with different initial conditions. Solid vertical lines indicate the interquartile range. Abbreviations: Stationary (`S`), Cylindrical (`C`) and Informative (`I`) covariances; Quadratic Mean (`+QM`); Origin (`+X0`) and Adaptive (`+XA`) anchors; acquisitions within Trust Region (`+TR`); Gaussian-Weighted Expected Improvement with fixed (`+GWEIX0`) and adaptive (`+GWEIXA`) location; Gaussian Kernel-weighted Expected Improvement with fixed (`+GKEIX0`) and adaptive (`+GKEIXA`) location.

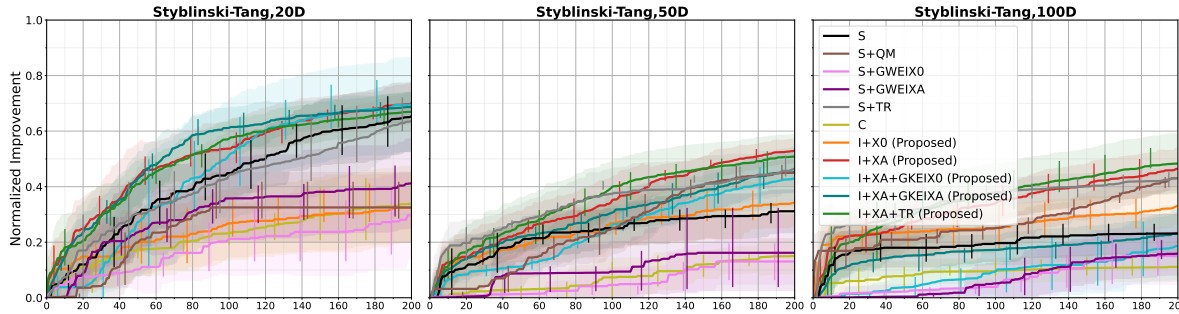

Figure 25: Performance on the Styblinski-Tang function with 20, 50 and 100 dimensions. Solid curves and shaded regions represent the mean and standard deviation of the normalized improvement, computed over 10 trials with different initial conditions. Solid vertical lines indicate the interquartile range. Abbreviations: Stationary (`S`), Cylindrical (`C`) and Informative (`I`) covariances; Quadratic Mean (`+QM`); Origin (`+X0`) and Adaptive (`+XA`) anchors; acquisitions within Trust Region (`+TR`); Gaussian-Weighted Expected Improvement with fixed (`+GWEIX0`) and adaptive (`+GWEIXA`) location; Gaussian Kernel-weighted Expected Improvement with fixed (`+GKEIX0`) and adaptive (`+GKEIXA`) location.

## J  High-dimensional Experiments with Larger Evaluation Budgets

Thus far, we have considered a budget of 200 acquisitions. As highlighted in Sections 2.1 and 3, the rationale is that BO is competitive when function evaluations are expensive, putting a practical constraint on the size of the budget, even for high-dimensional problems.[5] Additionally, we have favored an analysis in terms of sample efficiency, demonstrating that the proposed methodology, which employs informative covariance functions, offers methods that discover superior solutions with fewer function evaluations.

Despite the rationale above, we now increase the evaluation budget to 600 acquisitions. Figure 26 shows the results for two test functions that we consider representative (see Appendix B.3). The Rosenbrock function features a banana-shaped valley, with the global optimum relatively close to the center of the search space. Conversely, the Styblinski-Tang is a multimodal function, whose optimal solution is relatively distant from the center (local maximum). Once again, we observe that the proposed covariance functions can be combined with other methods, including quadratic mean functions and trust regions, to achieve higher sample efficiency and to discover superior solutions.

As a final note, while our methodology provides scalable methods in terms of dimensionality (e.g., `I+X0`, `I+XA`), these still share some of the limitations of standard GPs. In particular, when dealing with large training sets, standard GPs become impractical because they require $\mathcal{O}(n^3)$ computation and $\mathcal{O}(n^2)$ memory, where $n$ is the number of training points. To address this limitation, one option is to use sparse GPs with inducing variables, as proposed by e.g. Titsias (2009) and Hensman et al. (2013). This allows for a significant reduction in computational and memory requirements due to low-rank matrix approximations.

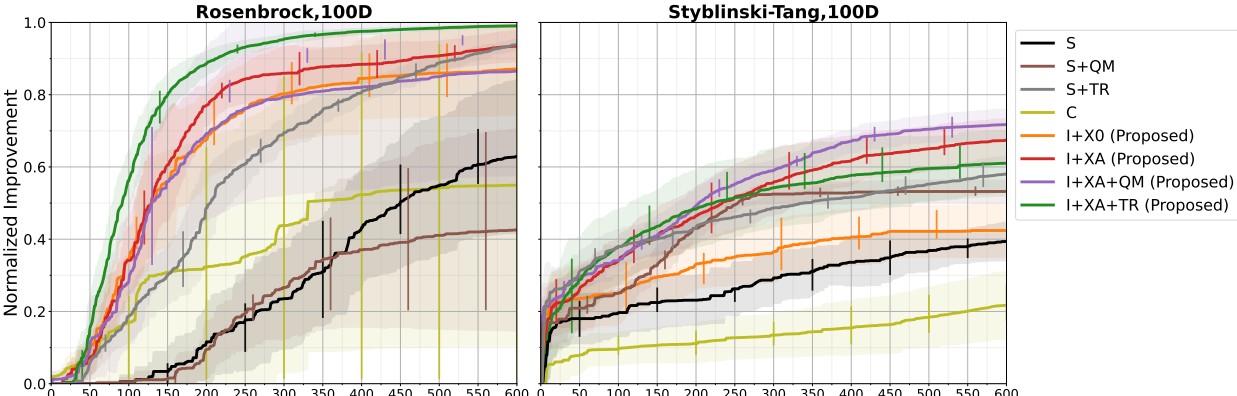

Figure 26: Performance on the 100-dimensional Rosenbrock and Styblinski-Tang functions. A value of 1 indicates that the global optimum has been found (zero simple regret). Solid curves and shaded regions represent the mean and standard deviation of the normalized improvement, computed over 10 trials with different initial conditions. Solid vertical lines indicate the interquartile range. Abbreviations: Stationary (`S`), Cylindrical (`C`) and Informative (`I`) covariances; Quadratic Mean (`+QM`); Origin (`+X0`) and Adaptive (`+XA`) anchors; acquisitions within Trust Region (`+TR`).

---

[5]Based on a similar reasoning, Eriksson & Jankowiak (2021) conducted 100-dimensional experiments with a maximum budget of 100 acquisitions.

