# OpenReview forum: "Bayesian Optimization with Informative Covariance"
_TMLR — Accepted by TMLR_

### Review · Reviewer_vdWc · 2023-02-09

**Summary Of Contributions:**

The paper investigates the strengths of using spatial-varying nonstationary covariance functions in Bayesian Optimization. The paper proposes a new class of covariance functions that leverage nonstationary to incorporate information about promising points in the search domain. The paper evaluates their proposed method on some high-dimensional test functions and real-world applications, under some prior information. It also shows that it can combine with other methods to improve its performance.

**Audience:**

Yes

**Claims And Evidence:**

No

**Requested Changes:**

+ More mathematical rigorous theorems to be included when analyzing the advantages of non-stationary covariance functions with spatially varying properties if this is listed as a contribution of the paper. Please see my comments in the previous section.
+ The proposed method needs to be elaborated more. For example, more discussion on the use of a set of multiple anchors, and more discussion regarding the design choices of the proposed method. The adaptive anchor strategy needs to be described in detail. Please see my comments in the previous section.
+ Experiments need to be improved. In particular, the number of iterations needs to be increased. The performance of all the methods with the standard regret also needs to be included in the experimental evaluation. The proposed methods to be used in synthetic and real-world experiments need to be consistent. Please see my comments in the previous section.

**Strengths And Weaknesses:**

Strengths:
+ The paper investigates an important and interesting problem, that is, using spatial-varying nonstationary covariance functions in Bayesian Optimization, which might help to improve the performance of BO in high-dimensional or complex optimization problems.
+ I like the idea of constructing spatial-varying nonstationary covariance functions by encoding the preferences for certain regions.
+ The descriptions of the experiments are detailed and thorough. Highly dimensional test functions are used to compare the performance between the proposed method and other baseline methods.

Weaknesses:

Even though I like the general idea of the paper, I found there are various issues with the current version of the paper:
+ One of the contributions listed by the paper is to provide a regret analysis that demonstrates the advantages of non-stationary covariance functions with spatially varying properties. However, Section 3 only provides some useful insights regarding the regrets of the spatial-varying nonstationary covariance functions, and there are no rigorous theorems proving that the spatial-varying nonstationary covariance function will tighten the regret bound. A lot of discussion/analysis in Section 3 should be formally formulated as theorems to guarantee their correctness. Lemma A.1 is not really helpful (1) it relies on an assumption that is not guaranteed to occur, (2) its conclusion does not help to understand the property of the regret bound as
+ The paper aims to provide a general formula for spatial-varying nonstationary covariance functions with a set of anchors, but it eventually only focuses on covariance functions with only one single anchor – which is very limited. And there are various design choices regarding the set of anchors (or the single anchor) that are not mentioned. For example, how is the distance function d_l chosen?  What is the range of w_l? Does it need to be larger than 1? How can we ensure \phi(x) to be positive? Why, in Eq. (11), x* is used, but not x? I feel really confused when reading the equation. What is the effect of choosing the wrong positions for the anchors? Currently, the experiment’s setup is that the global optimum is at the centre of the search space, and the anchor is chosen based on that. However, when we don’t know the location of the global optimum, how can we choose the anchors to ensure good performance?
+ The adaptive anchor strategy is not clearly described in the paper. There is only one sentence mentioning it in the experiment section (adaptive anchor section). It should be described clearly in Section 4 if it is part of the proposed method.
+ The experimental evaluation is quite limited, in my opinion. First, it is only conducted with one single anchor, which does not help to understand the generic method. Second, the number of iterations in each test case is too small (only 200). Normally for highly dimensional optimization problems, the number of iterations needs to be chosen much larger to understand the performance of the proposed method. Third, the normalized regret is used, which is a non-standard metric. Besides, in Section 3, the paper provides analysis regarding the standard regret, but in the evaluation, the normalized regret is used. Finally, the proposed methods used in the synthetic experiments are not the same as the proposed methods used in the real-world applications (in real-world experiments, there are I+XA+QM and I+XGood). The use of the proposed methods should be consistent in both synthetic and real-world experiments.

---

> ### Author Response · Authors · 2023-02-28
> **Response to Reviewer vdWc**
>
> Thank you for your review and valuable feedback. Please find below our answers to your questions and comments.
>
> >More mathematical rigorous theorems to be included when analyzing the advantages of non-stationary covariance functions with spatially varying properties if this is listed as a contribution of the paper.
>
> We have revised our list of contributions to reflect this point. In particular, we have removed this analysis from the list as it would require a more formal presentation. In the revised manuscript, this section is now introduced before listing the (other) contributions, providing the foundations for our methodology. This includes a discussion of the advantages and regret bounds for covariance functions with spatially-varying parameters, which we have not previously found in the literature.
>
> >Lemma A.1 is not really helpful (1) it relies on an assumption that is not guaranteed to occur, (2) its conclusion does not help to understand the property of the regret bound as
>
> 1. While it is true that the assumptions of Lemma A.1 are not guaranteed to occur in practice (as we discuss in Section 3), they are standard in the literature that we cite [e.g. 1, 2, 3]. In particular, the assumption that the interval includes the objective function $f$ follows from $f$ being assumed to live in a known RKHS for which [e.g. 4, Theorem 7] provides an expression for the factor $\beta$ that guarantees the condition holds with high probability, if the noise is sub-Gaussian. If the RKHS is not known, it is possible to follow an approach similar to that proposed by Berkenkamp et al. (2019) [5], by increasing the underlying function class over time (e.g., increasing the RKHS norm and decreasing the lengthscales).
> 2. The sentence by the reviewer seems to be incomplete. But in brief, Lemma A.1 allows us to associate wider intervals with larger regret bounds in Section 3, as part of our discussion of regret bounds.
>
> >focuses on covariance functions with only one single anchor
>
> >more discussion on the use of a set of multiple anchors
>
> With the revision, we now discuss the focus on a single anchor and possible options for covariance functions with multiple anchors in Section 5.4 (Limitations). However, as we make clear in that section, and partly in the Conclusion, a careful treatment of multiple anchors is out of scope for this paper and left for future work, requiring additional analysis and testing.
>
> >there are various design choices regarding the set of anchors (or the single anchor) that are not mentioned
>
> >how is the distance function d_l chosen?
>
> This is in fact discussed in Section 5.1 (and Appendices B and F that we reference in that section). In brief, we there state that "For black-box optimization, in which little is known in advance, a default kernel and distance function may be used, and their hyperparameters learned by empirical Bayes. To avoid overfitting and reduce computational complexity, regularizing hyperpriors and hyperparameter tying are effective techniques that may also be used. [...]" Please see that section for equations and more details (e.g., $d_0 = d_\mathrm{WE}$).
>
> >What is the range of w_l?  Does it need to be larger than 1? How can we ensure \phi(x) to be positive?
>
> To ensure that $\phi$ is non-negative, $w_l$ should be non-negative. This is pointed out  before we introduce Equation 11 (Section 4).  In practice, as shown in Equation 13, we use the reparametrization $r_l = 1/w_l \in (0, 1]$, which means $w_l \ge 1$, i.e., we increase the prior variance and shrink lengthscales as we move closer to anchors (see e.g. Figure 4).
>
> >Why, in Eq. (11), x* is used, but not x?
>
> We here follow the arguably not very precise but common practice to overload notation and use the argument of the pdf to specify it. As mentioned before introducing this equation, $p(\mathbf{x}^\star)$ is a "distribution over promising points", reflecting the prior belief about the possible locations of the optimum. We use $\mathbf{x}^\star$ consistently in Equation 11 and Equation 1, where it denotes the optimal solution.

---

> > ### Author Response · Authors · 2023-02-28
> > **Response to Reviewer vdWc (cont.)**
> >
> > >What is the effect of choosing the wrong positions for the anchors? Currently, the experiment’s setup is that the global optimum is at the centre of the search space, and the anchor is chosen based on that. However, when we don’t know the location of the global optimum, how can we choose the anchors to ensure good performance?
> >
> > In all our experiments, the anchor is misspecified to different degrees. In some test cases (e.g., QBranin, Rosenbrock), the anchor in I+X0 (center/origin) is close to the optimal solution. In other test cases (e.g. shifted QBranin/Rosenbrock, Styblinski-Tang), the optimal solution is relatively distant from the center. This is discussed in Section 5.2: "The assumption that optimal solutions are close to the center is strong. [...]  Despite the lower apparent vulnerability of I+X0 to anchor misspecifications, this motivates the use of informative covariance functions with an adaptive anchor (I+XA)."
> >
> > >The adaptive anchor strategy is not clearly described in the paper. There is only one sentence mentioning it in the experiment section (adaptive anchor section). It should be described clearly in Section 4 if it is part of the proposed method.
> >
> > In Section 4, we propose a methodology based on informative covariance functions (not a single method). In Section 5, we test several BO methods based on this methodology. A list of methods can be found in Appendix D. With the revision, we now introduce the methods at the beginning of Section 5 to increase clarity.
> >
> > In the adaptive case, the anchor is placed at the incumbent solution. Formally, $\mathbf{x}_0^{(n)} = \mathbf{x}_\mathrm{best}^{(n_0+n)}$ (as defined in Section 5.1), where the incumbent solution $\mathbf{x}_\mathrm{best}^{(n_0+n)}$ is the point in the evidence set with the lowest observed value after $n$ acquisitions (see also Appendix B.2). In Section 5.2, we added a sentence that clarifies this.
> >
> > >the number of iterations in each test case is too small (only 200). Normally for highly dimensional optimization problems, the number of iterations needs to be chosen much larger to understand the performance of the proposed method.
> >
> > In the revised document, Appendix J includes experiments with larger evaluation budgets. We there also clarify our rationale for focusing on restricted evaluation budgets. In our view, a basic motivation behind BO is the optimization of expensive functions. In practice, increasing the budget is often not feasible due to the computational complexity of standard GPs, which is cubic, and the fact that function evaluations can be time-consuming (e.g., days, weeks) or expensive in other ways, possibly incurring high monetary costs. Therefore, we believe that improving the performance (sample efficiency) of BO in the small sample regime is crucial. Recent studies, such as [6], have performed 100-dimensional experiments with a maximum of 100 acquisitions, based on similar reasoning.

---

> > > ### Author Response · Authors · 2023-02-28
> > > **Response to Reviewer vdWc (cont. 2)**
> > >
> > > >the normalized regret is used, which is a non-standard metric
> > >
> > > We use a normalized performance metric because standard (simple) regret depends on 1) initial conditions and 2) the range of the objective function. We ensure that the performance metric is invariant to these conditions, always starting from 0 and only attaining the value of 1 if a global optimum is found (zero simple regret). In our opinion, this allows for a more straightforward analysis. It can also be observed mathematically that a larger normalized improvement corresponds to lower regret. With the revision, we have included the above justification for our choice of metric. Importantly, our finding that the proposed class of GP priors can significantly increase sample efficiency of BO remains valid.
> > >
> > > >the proposed methods used in the synthetic experiments are not the same as the proposed methods used in the real-world applications (in real-world experiments, there are I+XA+QM and I+XGood). The use of the proposed methods should be consistent in both synthetic and real-world experiments.
> > >
> > > We agree that it may appear that there is an inconsistency in the use of methods. However, we have included I+XA+QM in all our experiments. We did not include all tested methods in Figure 5 (synthetic test functions) due to legibility concerns. Some figures were already quite dense as noted by reviewer 7J43 (but this has now been addressed, see our response to them).
> > >
> > > The results with I+XA+QM (and other methods) can be found in Appendix D (see our remark in the caption of Figure 5).  In particular, in Appendix D.2, we write that "in tests where S+QM performed comparatively well, e.g. Styblinski-Tang (Figure 13), the combination I+XA+QM proved to be effective", performing worse than I+XA otherwise.
> > >
> > > I+XGood is exclusive to MNISTW. In this application, S+QM performed well, and we wanted to show that by using stronger prior information (a solution found by S+QM) we can be more sample efficient. Unlike synthetic experiments, in these applications we don't know the objective function landscapes, including the locations of global optima. Finally, it is important to keep in mind that we are not advocating for a single method, but for a methodology based on GP priors with informative covariance functions (I), where we leverage second-order nonstationarity to incorporate information about promising points for optimization. This is possible without modifying the acquisition function.
> > >
> > > **References**
> > >
> > > [1] Niranjan Srinivas, et al., Gaussian Process Optimization in the Bandit Setting: No Regret and Experimental Design, ICML, 2010.
> > >
> > > [2] Ziyu Wang and Nando de Freitas, Theoretical Analysis of Bayesian Optimisation with Unknown Gaussian Process Hyper-Parameters, arXiv:1406.7758, 2014.
> > >
> > > [3] Sattar Vakiliet, et al., On Information Gain and Regret Bounds in Gaussian Process Bandit, AISTATS, 2021.
> > >
> > > [4] Sayak Ray Chowdhury and Aditya Gopalan, On Kernelized Multi-armed Bandits, ICML, 2017.
> > >
> > > [5] Felix Berkenkamp, et al., No-Regret Bayesian Optimization with Unknown Hyperparameters, JMLR, vol. 20, no. 50, 2019.
> > >
> > > [6] David Eriksson and Martin Jankowiak, High-Dimensional Bayesian Optimization with Sparse Axis-Aligned Subspaces, UAI, 2021.

---

### Review · Reviewer_DmEg · 2023-02-10

**Summary Of Contributions:**

This paper considers Bayesian optimization (B) with non-stationary kernels. Typically, stationary kernels are used in BO, but this work argues both empirically and theoretically that informative, non-stationary GPs can lead to improved sample efficiency. Specifically, this work introduces novel spatially-varying lengthscales and prior variance functions that can exploit prior knowledge about promising regions, provides a regret analysis, and demonstrates improved optimization performance on high-dimensional test problems.

**Audience:**

Yes

**Claims And Evidence:**

Yes

**Requested Changes:**

* [Important] Why are there only high-dimensional test functions? An analysis of the performance of this kernel in lower dimensional problems would strengthen the paper.
    * Since the evaluation is only on high-dim problems, how do other SOTA high-dimensional BO methods like SAASBO (Eriksson et al., 2021) compare?
* [Important] More compelling real world examples would drive home why this is important
    * Can the authors include a real world example a reasonable anchor point is known apriori?
    * The NN problem is not very compelling real world use case
* [Nice to have] How does performance vary with different nonstationary choices (e.g. no tying r,d,k)?
* [Nice to have] How does performance vary based on the number of anchor points?
* [Nice to have] What about vanilla input warping (Snoek et al., 2014)?

Additionally, the organization of the discussion of the results hinders readability.
* In the fixed anchor section, it would be good to discuss poor performance on Styblinski Tang, before getting to the adaptive anchor point technique. I can see that the paper aims to craft a story here, but as a reader, I was wondering why the poor performance of the fixed anchor method was not discussed in the fixed anchor section. Perhaps starting with an introduction to all methods and then discussion the results in the figures would make the work easier to follow.


Nit: Page 4: \citet at the end of Section 2.5

**Strengths And Weaknesses:**

Strengths

The writing is clear and articulate. The formulation of spatially varying prior variance and lengthscales given one or more anchor points is novel and intriguing. Although one might not always have prior knowledge of promising anchor points, a technique for adapting the anchor point as the current best incumbent is proposed and empirically validated. The regret analysis and discussion motivated the spatially varying kernel. Moreover the work highlights how the proposed kernel is complementary to prior-guided BO and trust region BO.


Weaknesses

The main weakness is the empirical evaluation (see requested changes for comments). Compelling real world examples would greatly strengthen the paper. Moreover, since the empirical evaluation exclusively considers high-dimensional problems, additional high-dimensional BO baselines are warranted. It would strengthen the paper to include lower dimensional test functions---at a minimum to provide evidence for why non-stationary kernels are more beneficial/important for high-dimensional problems. If this paper is solely focused on high-dimensional BO, I think that decision should be better motivated (at least empirically) and that focus on high-dimensional BO should made clear early in the paper (and potentially in the title).

---

> ### Author Response · Authors · 2023-02-28
> **Response to Reviewer DmEg**
>
> Thank you for your supportive review and helpful feedback. Please find below our answers to your questions and comments.
>
> >Why are there only high-dimensional test functions? An analysis of the performance of this kernel in lower dimensional problems would strengthen the paper.
>
> In Figure 1, we show the results on a 1-dimensional experiment, where we point out that even in low-dimensional objectives there are failure modes affecting standard BO with standard GP priors. However, as we make it clear from the beginning (Abstract and Section 1), we focus on high-dimensional domains, which are considered to be the more challenging cases that would benefit most from further research. With our paper, we aim to address some of the challenges in high-dimensional BO, including the boundary issue, as we explain in Sections 3 (High-dimensional domains) and 5.1 (Baselines).
>
> >Since the evaluation is only on high-dim problems, how do other SOTA high-dimensional BO methods like SAASBO (Eriksson et al., 2021) compare?
>
> While multiple approaches have been proposed for high-dimensional BO, these do not make the same assumptions as we do, and most are in fact complementary. For instance, we have shown that we can combine trust-region BO (SOTA) with the proposed informative GP priors.
>
> The SAAS hyperprior assumes that the objective function has low effective dimensionality, but our test functions do not meet this assumption. In the revised manuscript, we show that it is possible to combine our methodology with the SAAS prior for the lengthscales (see new results in Appendix H). In Section 6 (Related Work), we also point out that high-dimensional BO methods relying on additivity and low effective dimensionality are complementary.
>
> >Can the authors include a real world example a reasonable anchor point is known apriori?
>
> The rover trajectory planning problem has been used by multiple other recent studies and the point [x_start, 0, ..., 0, x_end] is a reasonable anchor that is known a priori to perform relatively well, according to the reward function in Equation 43 (Appendix G) and map layout (Figures 6 and 19).
>
> >NN problem is not very compelling real world use case
>
> We have tested the NN problem because it was proposed by Oh et al. (2018) [1] as an application. Baseline C uses their proposed covariance function.
>
> Please note that our paper is a methodological one. We have assessed the performance on test functions as well as two applications that are used in recent BO papers. We consider a "real" real-world application to be out of scope for this paper. Such applications require considerable engineering, and are not necessarily featured in other recent methodological studies.
>
> >discuss poor performance on Styblinski-Tang
>
> We have revised Section 5.2. In particular, we now highlight that the performance on the Styblinsky-Tang function is discussed in the following paragraphs, where we compare I+X0 and I+XA with other baselines.
>
> >How does performance vary with different nonstationary choices (e.g. no tying r,d,k)?
>
> In the paper, we have tested and included different nonstationary choices. For instance, in Appendix D, the method I+XA+F uses relatively short lengthscales to compute $d_0$ and a fixed $r_0$. In Appendix E, we include an ablation study where spatially-varying prior variances and lengthscales are evaluated separately. In Appendix F, we perform a sensitivity analysis where different hyperpriors for $r_0$ are tested. However, many other design choices are possible due to the flexibility of the proposed covariance functions.
>
> >How does performance vary based on the number of anchor points?
>
> In the revised manuscript, Section 5.4 (Limitations) now includes a discussion of covariance functions with multiple anchors, as requested by reviewer vdWc. In that section, we also note that model selection can determine the number and location of anchors automatically. However, it is worth to keep in mind that a prior with an anchor close to an optimum is likely to perform better than another with multiple distant anchors.
>
> >What about vanilla input warping (Snoek et al., 2014)?
>
> In Section 6 (Related Work), we point out that the method proposed by Oh et al. (2018) [1], denoted by baseline C in the paper, outperformed the input warping method by Snoek et al. (2014) [2] on several high-dimensional tests, resulting in our decision to only implement and test C. Additionally, vanilla input warping and C can be augmented with spatially-varying (co)variances and should not necessarily be viewed as alternatives. We do agree, however, that exploring this further as part of future work is worthwhile.
>
> **References**
>
> [1] ChangYong Oh, et al., BOCK: Bayesian Optimization with Cylindrical Kernels, ICML, 2019.
>
> [2] Jasper Snoek, et al., Input Warping for Bayesian Optimization of Non-stationary Functions, ICML, 2014.

---

### Review · Reviewer_7J43 · 2023-02-15

**Summary Of Contributions:**

In this paper, the authors introduce nonstationary GP priors useful for Bayesian optimization. In particular, the authors assume the existence of a set of anchors (most commonly L=1 anchor in the paper, which could be taken e.g. to be the current incumbent solution). The prior variance is then taken to be a function that smoothly inflates the correlation of two points when they are near an anchor, and the lengthscales are defined by an input warping (e.g. Snoek et al., 2014) that shrinks lengthscales in a neighborhood around the anchors (e.g., the incumbent). These factors lead to the influence of the anchors being greatly expanded, resulting in the optimization procedure being naturally encouraged to explore more closely around anchors.

**Audience:**

Yes

**Broader Impact Concerns:**

No real concerns here.

**Claims And Evidence:**

Yes

**Requested Changes:**

- The plots can become fairly illegible with as many as 5 of the authors own methods appearing in many of the result plots. The authors really should consider picking 1-2 instantiations of their method as exemplars, and then leave e.g. the comparison of XA to X0 to a separate ablation study. In half of these plots, I can't tell if the error bars e.g. between I+XA and S+QM overlap because there are 4 other instances of their own method on the plot.

- The evaluation budget considered in every task is quite small. This is fine, because BayesOpt often targets the very low evaluation budget regime. However, I still think there's substantial value in at least *showing* us results as an ablation study with larger budgets on a few of these tasks. For example, S+TR does extremely poorly by N=200 observations in the authors' results on Rover, but obviously does fairly well on this task by N=7500 (Eriksson et al., 2018). Currently, the long term trade-offs of the authors' method is a fairly open question, and it's not obvious to me that the authors' method should have any scalability concerns, so the lack of these results is confusing to me.

- [Minor] I'd like the authors to add a piece of clarification about the prior variance function. In particular, it looks to me as though the prior covariance function in (12) will inflate the covariance between two points that are close to anchors, *regardless* of whether they are close to the *same* anchor. Trivially this is a nonissue in the case of one anchor (equation (13)), which the authors primarily focus on, but I'm curious whether this implies that, if anchors aren't chosen to have very similar function values (e.g., if the second best observation in the domain far from the first best happens to be much worse), this might lead to somewhat strange model fits.


**Strengths And Weaknesses:**

Overall, I think the authors' idea is great. This seems like a pretty nice way to get TR-like optimization behavior without necessarily having to use a TR (although the methods are complementary, as the authors point out), and in several experiments this seems to pan out in terms of getting fairly good optimization performance at lower evaluation budgets than we might expect using a TR. I especially like that a lot of the resulting parameters end up being learnable, which perhaps makes the authors' method require less hand crafting than some of the TR hyperparameters, which admittedly have reasonable out of the box settings.

On the weaknesses side, I didn't find the discussion around regret to be particularly useful. Some of the earliest approaches to provide regret bounds for Bayesian optimization in the setting where the hyperparameters aren't known in advance already view the objective function in that setting as drawn using a set of RKHS's indexed by a sequence of lengthscales, (e.g. Bull et al), and I think it's reasonably well known that the resulting regret bounds have to be pessimistic by analyzing under the shortest possible lengthscales. I understand that this section primarily serves as motivation for the authors' practical method, but the first bullet point at the end of Section 1 is perhaps a bit overstated as a result.

---

> ### Author Response · Authors · 2023-02-28
> **Response to Reviewer 7J43**
>
> Thank you for your encouraging review and constructive feedback. Please find below our answers to your questions and comments.
>
> >first bullet point at the end of Section 1 is perhaps a bit overstated
>
> With the revision, we have removed this point from the list of contributions. See also our response to reviewer vdWc.
>
> >The plots can become fairly illegible
>
> In the revised manuscript, we have made modifications to the figures in Section 5. In particular, we have increased the size of Figures 5, 6 and 7 (new). We also removed I+XA+QM from Figure 6 (now only shown in Appendix G) and we have split the single plot for MNISTW into two (Figure 7). While it may be difficult to distinguish the shaded regions (mean$\pm$ standard deviation) due to overlap, we believe that the error bars, which indicate the interquartile range, are now more discernible.
>
> >The evaluation budget considered in every task is quite small [...] it's not obvious to me that the authors' method should have any scalability concerns, so the lack of these results is confusing to me.
>
> With the revision, we have included additional experiments with larger evaluation budgets (Appendix J). As discussed in that section, our methodology can provide scalable methods in terms of dimensionality, but these still share some of the limitations of standard GPs. For instance, the inversion of the Gram matrix becomes an issue when the number of training points is large, but existing methods, e.g. based on inducing points, can be used to alleviate these computational issues.
>
> Regarding the rover trajectory planning problem, please see our discussion in Appendix G (see also Figure 20, OversmoothedRoverT vs RoverT).
>
> >the prior covariance function in (12) will inflate the covariance between two points that are close to anchors, regardless of whether they are close to the same anchor
>
> We thank the reviewer for the interesting question. While this is true, the covariance also depends on the distance between these two points because the informative covariance function (Equation 14) is given by the product of the prior covariance function and an input-warped stationary covariance function. As there are various possibilities to consider (e.g., hyperparameters that define distances, number and locations of anchors and training points), we prefer to avoid making general statements in this regard. In the revised document, we discuss in Section 5.4 (Limitations) that further research is needed for covariance functions with multiple anchors, which we leave for the future.

---

### Author Response · Authors · 2023-02-28
**Overview of the Major Changes**

We thank the reviewers for their comments. We have updated our manuscript to address their questions. In brief, the major changes are:

1. Updated the list of contributions (Section 1, Introduction)
2. Improved legibility of the plots (Section 5, Experiments)
3. Added an overview of the BO methods that use the proposed covariance functions (Section 5.1, Setting)
4. Added clarification about the normalized performance metric and its relation to simple regret (Section 5.1, Setting)
5. Added discussion about options and the additional work needed for covariance functions with multiple anchors (Section 5.4, Limitations)
6. Additional experiments with sparsity-inducing hyperpriors (Appendix H, SAAS)
7. Additional experiments with larger evaluation budgets (Appendix J)

In the revised document, these changes are marked in red.

---

### Decision · Action_Editors · 2023-03-18

**Recommendation:** Accept with minor revision

**Comment:**

Based on the reasons above, it's not an easy decision. I'll just recap the main points: (+) useful idea, that works in practice, but (-) some (not particularly challenging) experiments are missing. All reviewers are hesitating, 1 recommends to reject [vdWc: "My opinion towards the paper is quite marginal (...) which is not enough to convince me towards the acceptance of the paper"] while 2 think that the pros outweigh the cons [7J43: "I'm leaning towards acceptance here (...) I think the actual method here is actually pretty sensible, and the kind of thing you might actually try adding to your bayesopt code in practice with pretty minimal overhead. To me, that makes it a lot harder to outright reject the paper. Honestly, if they'd just come back and run every result with budgets up to a couple thousand, I'd be championing this paper strongly at this point"; DmEg: "I lean slight toward acceptance, but I acknowledge the criticisms raised by Reviewer vdWc and I would not oppose a rejection", "Despite that, I do think this work would be of interest to the community and that its merits outweigh its weaknesses"]

Because I agree that this method should be in the toolbox of practitioners of Bayesian optimization, I will follow the opinion of Reviewers 7J43 and DmEg and recommend to accept the paper. We do not completely understand why the authors did not add more large budget experiments, this would have made the experimental section much stronger at little cost. I will recommend "accepted under minor revision" to allow the authors to include more experiments. Of course, at that stage, there won't be any more reviews, but I insist that the authors decision to include more experiments could make their own work more convincing. It's their choice ultimately -- on our side, we believe it is a useful contribution, and as such it deserves to be published.

**Audience:**

Anyone working on Bayesian optimization (theory and practice).

**Claims And Evidence:**

This paper proposes new nonstationary covariance functions in order to incorporate information about promising points (anchors) in Bayesian optimization. The paper starts with a regret analysis of Bayesian optimization that emphasizes the importance of nonstationary covariance functions. Then, the new covariance is introduced, and tested in high-dimensional problems. The improvement over stationary function is clear in the experiments.

The referees are unanimous on the following points (and I totally agree with them):

1) they like the idea proposed by the authors, and acknowledge its good empirical performances [DmEg: "The formulation of spatially varying prior variance and lengthscales given one or more anchor points is novel and intriguing"; "I find the main contribution of the paper to be a technique that works well empirically on a variety of benchmarks"; vdWc: "I like the idea of constructing spatial-varying nonstationary covariance functions by encoding the preferences for certain regions"; 7J43: "Overall, I think the authors' idea is great"]

2) they believe that the regret analysis is not very informative nor 100% rigourous [vdWc: "However, Section 3 only provides some useful insights regarding the regrets of the spatial-varying nonstationary covariance functions, and there are no rigorous theorems proving that the spatial-varying nonstationary covariance function will tighten the regret bound"; 7J43: "On the weaknesses side, I didn't find the discussion around regret to be particularly useful", "the theory here is pretty meaningless"]. However, they acknowledge it can be kept as a motivation for the method proposed by the authors [7J43: "I understand that this section primarily serves as motivation for the authors' practical method, but the first bullet point at the end of Section 1 is perhaps a bit overstated as a result"; DmEg: "Although the regret analysis is not particularly insightful, it serves more as motivation for the method than the main contribution"]

3) finally, while the experiments are convincing, they are all in the small budget regime [7J43: " the empirical study has a few flaws including the lack of any large budget examples"; vdWc: "The main weakness here is that the evaluation budget for the experiments is quite small (it's much smaller than the standard evaluation budget in BO papers)"]
Following this comment, the authors included two experiments with larger budget. The reviewers think this improves the paper, even though they expected more: [DmEg: "I appreciate the empirical results that the authors added (...) however, the authors did not reproduce a compelling real world example is where this is useful."; vdWc: "The authors' response demonstrate some new results with longer budget, but only on 2 synthetic problems - which is not enough to convince me towards the acceptance of the paper"; 7J43: "I still think it's a little silly to only evaluate the functions in the paper up to budgets of like 200 -- we'll just never know how they do past that unless we run it ourselves. I appreciate that they've added a couple functions with higher budgets, but I'm still baffled why this isn't just done on every single result they ran -- we're talking about a handful of hours of compute time for the entire paper here"]